# High-content synaptic phenotyping in human cellular models reveals a role for BET proteins in synapse assembly

**Martin H Berryer[1,2], Gizem Rizki[1,2], Anna Nathanson[1,2], Jenny A Klein[1,2], Darina Trendafilova[1,2], Sara G Susco[1,2], Daisy Lam[1], Angelica Messana[1], Kristina M Holton[1,2], Kyle W Karhohs[3], Beth A Cimini[3], Kathleen Pfaff[2], Anne E Carpenter[3], Lee L Rubin[1,2], Lindy E Barrett[1,2]***

[1]Stanley Center for Psychiatric Research, Broad Institute of MIT and Harvard, Cambridge, United States; [2]Department of Stem Cell and Regenerative Biology, Harvard University, Cambridge, United States; [3]Imaging Platform, Broad Institute of MIT and Harvard, Cambridge, United States

**Abstract** Resolving fundamental molecular and functional processes underlying human synaptic development is crucial for understanding normal brain function as well as dysfunction in disease. Based upon increasing evidence of species-divergent features of brain cell types, coupled with emerging studies of complex human disease genetics, we developed the first automated and quantitative high-content synaptic phenotyping platform using human neurons and astrocytes. To establish the robustness of our platform, we screened the effects of 376 small molecules on presynaptic density, neurite outgrowth, and cell viability, validating six small molecules that specifically enhanced human presynaptic density in vitro. Astrocytes were essential for mediating the effects of all six small molecules, underscoring the relevance of non-cell-autonomous factors in synapse assembly and their importance in synaptic screening applications. Bromodomain and extraterminal (BET) inhibitors emerged as the most prominent hit class and global transcriptional analyses using multiple BET inhibitors confirmed upregulation of synaptic gene expression. Through these analyses, we demonstrate the robustness of our automated screening platform for identifying potent synaptic modulators, which can be further leveraged for scaled analyses of human synaptic mechanisms and drug discovery efforts.

## Editor's evaluation

Berryer et al. report on an automated and quantitative platform to study the number of synaptic inputs formed in networks of human excitatory neurons and astrocytes in vitro. The utility of the platform was tested by screening a large collection of small molecules; several modulators of synapse density were identified and validated in follow-up experiments. The automated platform substantially extends what is currently available, particularly with respect to the automation of the initial analysis steps. The positive hits identified here, the inhibitors of bromodomain and extraterminal (BET) family of gene expression regulators, are important, and will likely contribute to the understanding of the mechanisms of human synapse assembly.

*For correspondence:
lbarrett@broadinstitute.org

## Introduction

Synapses are intercellular junctions between neurons crucial for information processing. Seminal studies using the frog neuromuscular junction (*del Castillo and Katz, 1954*) or giant squid synapse

(*Bullock and Hagiwara, 1957*) established fundamental principles of chemical transmission. Subsequent genetic and neurobiological studies have further illuminated our understanding of the mammalian central nervous system and have implicated synaptic alterations in brain disorders, including autism spectrum disorder (*Bourgeron, 2015*; *De Rubeis et al., 2014*), schizophrenia (*Fromer et al., 2014*; *Kirov et al., 2012*; *Purcell et al., 2014*), and Alzheimer's disease (*Kunkle et al., 2019*; *Spires-Jones and Hyman, 2014*). Thus, insights into synaptic development and function are critical for our understanding of normal brain function as well as dysfunction in disease. Diverse genes such as neuroligins and neurexins (*Graf et al., 2004*; *Varoqueaux et al., 2006*), neurotrophic receptors (*Elmariah et al., 2004*), EphrinB and EphrinB receptors (*Henderson and Dalva, 2018*), and cadherins, protocadherins, and catenins (*Arikkath and Reichardt, 2008*; *Weiner et al., 2005*; *Yamagata et al., 2018*) have been implicated in synaptic biology, although given the complex nature of mammalian brain development and function, molecular programs underlying neuronal connectivity have yet to be fully resolved.

Notably, most synaptic studies in mammals use rodent primary neuronal cultures (*Linhoff et al., 2009*; *Nieland et al., 2014*; *Sharma et al., 2013*; *Sharma et al., 2012*; *Verstraelen et al., 2014*). While the general principles of neuronal and synaptic development are thought to be conserved across species (*DeFelipe, 2011*), increasing evidence suggests that species-divergent features influence signal processing in dendrites and axons, which in turn may contribute to differences in cognition and function across organisms (*Eyal et al., 2016*; *Molnár et al., 2016*). Indeed, it is possible that human-specific aspects of synapse biology contribute to higher order cognition and function and may be perturbed in disease, underscoring the importance of utilizing both highly tractable, well-studied animal models as well as emerging human cellular in vitro models to identify and analyze convergent and divergent principles. In this regard, studies comparing human and rodent cortical neurons have revealed differences in membrane capacitance (*Eyal et al., 2016*), epigenetic signatures (*Luo et al., 2017*), gene expression and transcriptional regulatory mechanisms (*Bakken et al., 2016*; *Hodge et al., 2019*; *Zeng et al., 2012*), as well as changes in activity-dependent gene expression (*Pruunsild et al., 2017*) associated with distinct somatic, axonal, dendritic, and spine morphologies (*Beaulieu-Laroche et al., 2018*; *Benavides-Piccione et al., 2002*; *Mohan et al., 2015*), which correlate with differences in synaptic plasticity (*Szegedi et al., 2016*; *Verhoog et al., 2013*) and likely contribute to the differences observed in the integrative properties of the neuron (*Molnár et al., 2016*; *Szegedi et al., 2016*; *Verhoog et al., 2013*). For example, electrophysiological analyses have revealed key differences in spike timing-dependent plasticity in the human cortex compared with rodent, including the ability of human synapses to change strength bidirectionally in response to spike timing (*Verhoog et al., 2013*). A study of connections between human pyramidal neurons and fast-spiking GABAergic interneurons reported four times more functional presynaptic release sites in humans compared with rodents (*Molnár et al., 2016*). Paralleling functional differences, dendritic spines from pyramidal neurons in the human brain were reported to be larger, longer, and more densely packed compared with their rodent counterparts (*Benavides-Piccione et al., 2002*). Moreover, astrocytes are known to play critical roles in synapse development, maintenance, and function (*Allen et al., 2012*; *Allen and Eroglu, 2017*; *Bernardinelli et al., 2014*; *Di Castro et al., 2011*; *Krencik et al., 2017*; *Panatier et al., 2011*; *Ullian et al., 2001*), and human astrocytes exhibit transcriptomic profiles as well as functional and structural properties that diverge from rodent astrocytes (*Hawrylycz et al., 2012*; *Oberheim et al., 2009*). Thus, human and rodent astrocytes likely differ in their contributions to synaptic and neuronal function (*Oberheim et al., 2009*; *Zhang et al., 2016*). The above species-specific features highlight the importance of studying human neurons and human astrocytes to decode human nervous system function, which critically depends on networked neural activity involving both cell types.

Recent advances in human pluripotent stem cell (hPSC) biology and differentiation now allow for the generation of multiple human brain cell types in vitro including glutamatergic neurons (*Busskamp et al., 2014*; *Chambers et al., 2009*; *Zhang et al., 2013*). This has provided key insights into molecular mechanisms underlying normal synaptic development (*Patzke et al., 2019*), as well as alterations in human neurodevelopmental and neurological disorders such as autism (*Krey et al., 2013*; *Marchetto et al., 2017*; *Yi et al., 2016*), schizophrenia (*Pak et al., 2015*; *Yoon et al., 2014*), Alzheimer's disease (*Israel et al., 2012*; *Lin et al., 2010*), and Parkinson's disease (*Sánchez-Danés et al., 2012*; *Vera et al., 2016*). Advances in differentiation technologies have also facilitated the scalable production of hPSC-derived neurons to allow for phenotypic cell-based screening assays. Both genetic and pharmacological drug screening strategies using hPSC-derived neurons have been employed to study

pathways regulating Tau and β-amyloid production in Alzheimer's disease (*Brownjohn et al., 2017*; *Kondo et al., 2017*; *van der Kant et al., 2019*; *Wang et al., 2017*). Other studies have used hPSC-derived neurons to measure neurotoxicity (*Ryan et al., 2016*), neurite outgrowth (*Sherman and Bang, 2018*), and gene function using CRISPR interference and activation (*Tian et al., 2021*; *Tian et al., 2019*), further supporting the utility of such models for high-throughput screens. However, despite their unprecedented potential, scaled analyses of synaptic mechanisms using human cellular systems have not yet been achieved, in part due to the technical challenges associated with generating and analyzing reproducible culture preparations essential for these analyses. Indeed, while studies using human neurons have quantified the effects of candidate small molecules on synaptic density (*Green et al., 2019*; *Patzke et al., 2019*; *Trujillo et al., 2021*), to date we are not aware of any automated systems for their unbiased interrogation. This will become increasingly relevant for the study of basic synaptic mechanisms as additional species divergent features of brain cell types are identified and for the study of complex human diseases that impact synaptic function.

We therefore devised a fully automated and quantitative high-content synaptic phenotyping platform using in vitro-derived human glutamatergic neurons and primary human astrocytes. To demonstrate the utility of our platform, we screened effects of 376 small molecules on presynaptic density, neurite outgrowth, and cell viability. These analyses identified six small molecules that specifically enhanced presynaptic density. We then probed the most prominent hit class, bromodomain and extraterminal (BET) inhibitors, through global transcriptional analyses. While BET inhibitors have been associated with decreased synaptic gene expression in rodent models (*Korb et al., 2015*; *Sullivan et al., 2015*), we confirmed upregulation of synaptic gene expression in human cellular models, supporting the results of our synaptic assay. Taken together, we have established a novel human synaptic phenotyping platform that effectively identifies synaptic modulators in vitro. This platform can now be leveraged for further interrogation of human synaptic mechanisms and drug discovery efforts.

## Results
### Generation of robust human neuronal cultures for synaptic screening applications

To reproducibly generate large batches of human-induced neurons (hNs) from hPSCs, we used a well-described in vitro differentiation protocol based on ectopic expression of Neurogenin 2 (NGN2) combined with developmental patterning using small molecules (*Nehme et al., 2018*; *Zhang et al., 2013*). We selected this protocol based on extensive molecular and physiological characterization of these hNs alongside their use in multiple studies of disease-associated genes (*Busskamp et al., 2014*; *Deneault et al., 2018*; *Lin et al., 2018*; *Meijer et al., 2019*; *Nehme et al., 2018*; *Pak et al., 2015*; *Yi et al., 2016*; *Zhang et al., 2013*; *Zhang et al., 2018*) and in the only published genome-wide CRISPRi/CRISPRa screens of in vitro-derived human neurons to date (*Tian et al., 2021*; *Tian et al., 2019*). Importantly, studies using these hNs have identified physiological and/or morphological phenotypes upon perturbation of synaptic genes (e.g., *NRXN1*, *SHANK3*), which have then been recapitulated using other differentiation paradigms, further validating the accuracy of synaptic phenotypes detected in hNs generated with this protocol (*Pak et al., 2015*; *Yi et al., 2016*). Moreover, most neuronal differentiation paradigms in vitro generate heterogeneous cell types at lower throughput (*Chambers et al., 2009*) however, this protocol produces relatively homogeneous populations of glutamatergic neurons at large scale, making it uniquely suited for screening applications (*Nehme et al., 2018*; *Tian et al., 2021*; *Tian et al., 2019*). Data from our laboratory and others confirm that hNs begin to show electrophysiological activity around day 21 in vitro displaying spontaneous excitatory post-synaptic currents (sEPSCs) and NMDAR-mediated currents (*Meijer et al., 2019*; *Nehme et al., 2018*; *Susco et al., 2020*). However, it is important to note that at this timepoint synapses and synaptic network are still maturing.

To overcome the limitations of lentiviral titer used to transduce NGN2 and to enhance differentiation efficiency, we used a pair of TALENs to stably introduce a doxycycline-inducible NGN2 cassette (*Zhang et al., 2013*) into the AAVS1 safe harbor locus of the hPSC line H1 to generate a zeocin-resistant doxycycline-inducible NGN2 hPSC line, referred to as iNGN2-hPSC (*Figure 1a*), similar to the approach used by *Meijer et al., 2019*. Upon transfection, cells were selected for geneticin resistance and individual clones were isolated, expanded, re-plated for genomic DNA extraction and PCR

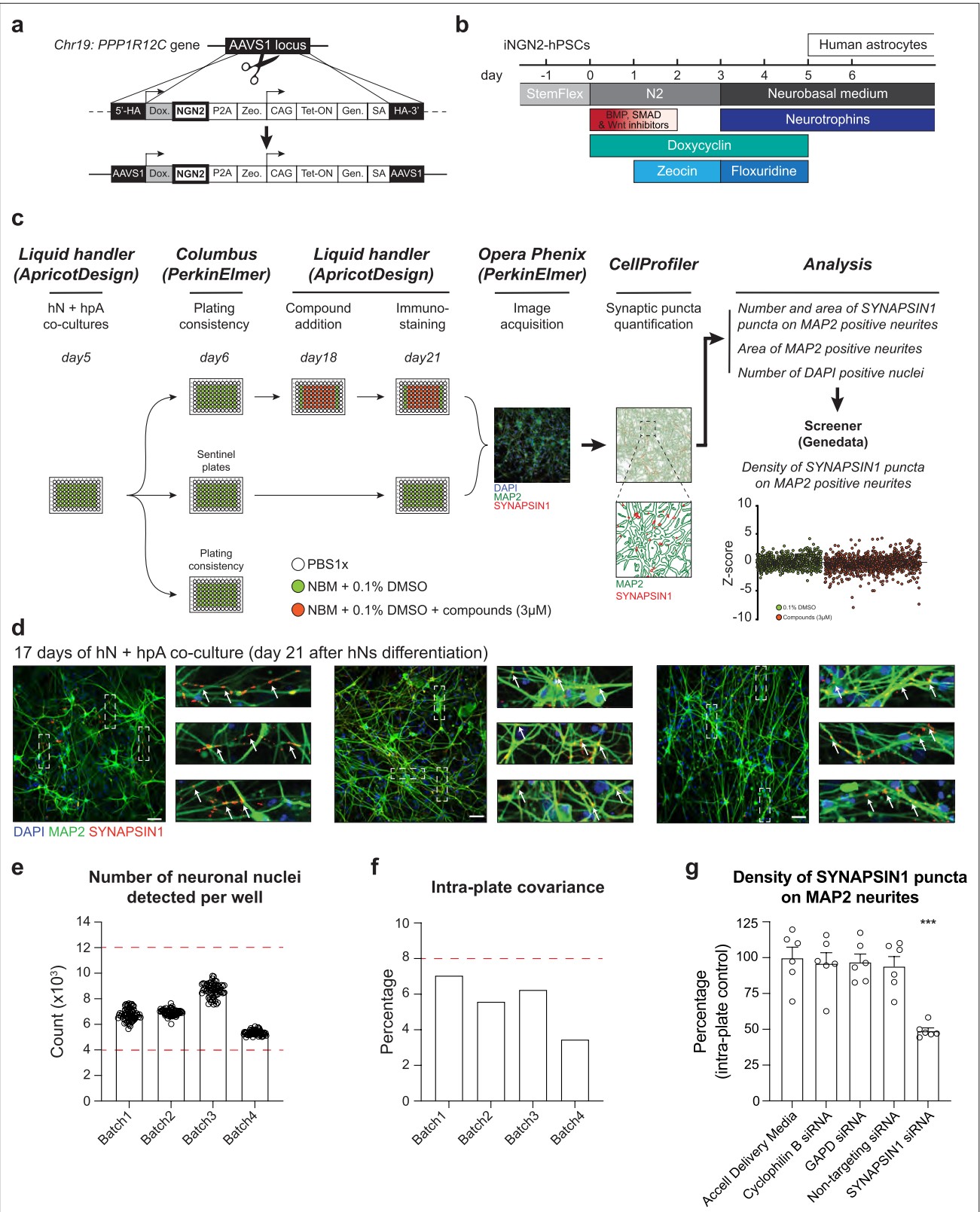

**Figure 1.** Development of an automated, high-content human synaptic phenotyping platform. (**a**) Schematic of stable integration of a doxycycline-responsive NGN2 (iNGN2) cassette into human pluripotent stem cells (hPSCs). TALENs were used to insert iNGN2 into the AAVS1 safe harbor locus of the *PPP1R12C* gene. (**b**) Summary of human-induced neuron (hN) protocol. hPSCs are differentiated into neural progenitor like cells using extrinsic inhibition of BMP, SMAD, and Wnt in combination with doxycycline-driven transient ectopic NGN2 expression. Zeocin is used as a selective agent.

*Figure 1 continued on next page*

*Figure 1 continued*

Neural progenitor-like cells were then incubated with neurotrophins in addition to the anti-mitotic agent floxuridine. hNs are then maintained in neurobasal medium and neurotrophins. (**c**) Flowchart of the high-content synaptic screening platform. One day after automated seeding of hNs and hpAs (using a liquid handler, day 5 of hN differentiation), a random plate is selected for assessing plating consistency (Columbus script). On day 18, treatments (shown here as small molecules at 3 µM) are administered in triplicate for 72 hr using a liquid handler; two plates are used as sentinel or reference. On day 21, co-cultures are then stained for synaptic markers using the liquid handler and high-content images are acquired using the Opera Phenix (Perkin Elmer). Data are then processed through CellProfiler and Screener to quantify the number and area of SYNAPSIN1 puncta on MAP2-expressing neurites, the area of MAP2-positive neurites, the number of DAPI-positive nuclei, and the density of SYNAPSIN1 puncta on MAP2-expressing neurites (Z-score). (**d**) Representative images of hNs co-cultured with hpAs and stained for SYNAPSIN1 (red) and MAP2 (green). Cells are counterstained with DAPI (blue). Scale: 100 pixels. Insets: arrows show SYNAPSIN1 puncta localized on MAP2-positive neurites. (**e, f**) Acceptance criteria for plating consistency. Plating consistency for each batch is determined using a Columbus script by quantifying detected hN nuclei per well (**e**) as well as the intra-plate covariance (**f**) of the randomly selected plate on day 6 after hN differentiation. Circles represented the number of neuronal nuclei detected per well from the plates randomly selected on day 6 (one day after seeding the co-culture). Thresholds for inclusion (red dashed lines) are set as above 4000 detected hN nuclei per well and below 12,000 detected hN nuclei per well (**e**) and a covariance below 8% across the plate (**f**). n = 1 x 96 well-plate per Batch which corresponds to 60 wells per 96-well plate per Batch. Error bars are shown as mean +/- SEM. (**g**) Quantification of the density of SYNAPSIN1 puncta on MAP2 neurites after incubation with SYNAPSIN siRNA versus control conditions. n = 6 wells for each condition; ***p<0.0001, ANOVA with Dunnett's post hoc test. Error bars are shown as SEM.

The online version of this article includes the following figure supplement(s) for figure 1:

**Figure supplement 1.** Validation of stably integrated iNGN2 cassette in human pluripotent stem cells (hPSCs) and additional synaptic markers.

**Figure supplement 2.** Columbus script to evaluate acceptance criteria for each batch of cultures.

**Figure supplement 3.** Overview of automated layered profiling and quantitative analysis of SYNAPSIN1 puncta on MAP2 neurites (ALPAQAS).

**Figure supplement 4.** ALPAQAS detects SYNAPSIN1 knock-down.

**Figure supplement 5.** Workflow supported by Genedata.

analysis of the transgene integration, and further analyzed by G-band karyotyping (*Figure 1—figure supplement 1*). To generate large-scale neuronal preparations essential for screening applications, we then differentiated the iNGN2-hPSCs into hNs as described (*Nehme et al., 2018*; *Figure 1b*).

## Establishment of automated pipelines for high-content human presynaptic phenotyping

We next developed a novel and scalable platform for automated quantification of presynaptic puncta in hNs co-cultured with human primary astrocytes (hpAs, ScienCell) (*Figure 1c*). First, because the number and distribution of synapses in a network is critically dependent upon neuronal density (*Cullen et al., 2010*), we used an automated liquid handling system (Personal Pipettor, Apricot Designs) to maximize pipetting precision and accuracy and reliably dispense batches of post-mitotic hNs and hpAs across the 60 inner wells of each 96-well plate. Upon hN co-culture with hpAs, we observed a dense synaptic network (*Figure 1d*). We specifically focused on the presynaptic marker SYNAPSIN1, which is crucial for maintenance, translocation, and exocytosis of synaptic vesicle pools, and the cyto-skeletal protein MAP2 (microtubule-associated protein 2), which is primarily enriched in perikarya and dendrites (*Caceres et al., 1984*) in addition to DAPI-expressing nuclei. We then quantified presynaptic puncta, defined as SYNAPSIN1 puncta assembled on MAP2-expressing neuronal dendrites, which are (1) robustly captured in our human cellular models; (2) essential for normal neuronal development with perturbations implicated in disease (*Südhof, 2017*; *Waites and Garner, 2011*); (3) a prerequisite for the assembly of postsynaptic machinery (*Friedman et al., 2000*; *Rohrbough et al., 2007*; *Sanes and Lichtman, 2001*); and (4) successfully measured in multiple synaptic screens performed in mouse (*Hempel et al., 2011*; *Spicer et al., 2018*), as well as in small-scale synaptic measurements performed using human cellular systems (*Chanda et al., 2019*; *Pak et al., 2015*; *Yi et al., 2016*). SYNAPSIN1 appeared in a discrete punctate staining pattern localized on and along the clearly defined MAP2-expressing neurites (*Figure 1d*). To enhance the resolution of presynaptic puncta and reduce nonspecific background for high-content imaging, we included glycine and TrueBlack Lipofuscin, key reagents for unmasking epitopes and quenching autofluorescence in our immunocytochemistry protocol.

In addition to quantifying SYNAPSIN1 on MAP2, we also assessed the possibility of incorporating an additional synaptic marker into our assay, testing conditions for labeling two additional presynaptic proteins: synaptophysin, SV2A, and four additional postsynaptic proteins: PSD-95, NLGN4, BAIAP2, and Homer1 (*Figure 1—figure supplement 1* and data not shown). Among the tested antibodies,

we were able to quantify signal for synaptophysin, involved in synaptic vesicle endocytosis, and PSD-95, a key postsynaptic scaffolding protein (*Figure 1—figure supplement 1*). Roughly half of the SYNAPSIN1 signal on MAP2 colocalized with synaptophysin and vice versa (*Figure 1—figure supplement 1*). Interestingly, studies in rodent cortex report that SYNAPSIN1 expression occurs prior to synaptophysin, potentially reflecting maturation of vesicle exocytosis mechanisms prior to endocytosis mechanisms (*Pinto et al., 2013*), which may contribute to the partial overlap in our assay. In general, SYNAPSIN1 is reported to be a more robust and specific marker for presynaptic terminals, with reports of synaptophysin also expressed at extrasynaptic sites (*Micheva et al., 2010*; *Pinto et al., 2013*). Indeed, the signal we obtained for synaptophysin was weak compared with the signal we obtained for SYNAPSIN1 (*Figure 1—figure supplement 1*). We observed moderate colocalization of PSD-95 with SYNAPSIN1; of the PSD-95 puncta on MAP2, 43.1% also colocalized with SYNAPSIN1 and of the SYNAPSIN1 puncta on MAP2, 28.8% colocalized with PSD-95 (*Figure 1—figure supplement 1*). This may indicate presynaptic assembly occurring prior to postsynaptic assembly (*Friedman et al., 2000*; *Rohrbough et al., 2007*; *Sanes and Lichtman, 2001*) at this early stage of neuronal development. Again, at this developmental timepoint, the signal obtained for PSD-95 was weak and of insufficient quality as well as consistency for screening applications (*Figure 1—figure supplement 1*). It was therefore not feasible to incorporate these additional synaptic markers into our high-content assay; thus, we focused on SYNAPSIN1 puncta assembled on MAP2-expressing neurites.

To rigorously confirm hN plating consistency (1 d after co-culture, which corresponds to 6 d after the start of hN differentiation), we developed an in silico script using Columbus Acapella software (PerkinElmer; *Figure 1—figure supplement 2*), quantified DAPI-expressing hN nuclei in the co-culture and established key thresholding criteria (i.e., the minimum number of detected hN nuclei per well and the related intra-plate covariance) for each batch of plates (*Figure 1e and f*). Of note, while several synaptic screens using mouse primary neurons report plating a range of 10,000–20,000 neurons per well in 96-well plates (*Nieland et al., 2014*; *Sharma et al., 2012*), we validated the number of hNs from a randomly selected plate 1 d after co-culture and found that 4000 surviving hNs per well was a minimum to establish a synaptic network and 12,000 surviving hNs per well was the maximum to circumvent hN clumping as well as reliably discriminate distinct SYNAPSIN1 puncta and track MAP2-expressing neurites for image processing (*Figure 1e*). Importantly, consistency of plating was high within batches, with an 8% covariance threshold (*Figure 1f*). Each treatment condition was compared to intra-plate/intra-batch controls and independent batches of plates were used as experimental replicates where treatment conditions were again compared to intra-plate/intra-batch controls.

Second, we developed an automated synaptic quantification pipeline called ALPAQAS (Automated Layered Profiling And Quantitative Analysis of Synaptogenesis) based on image analysis algorithms from the open-source software CellProfiler (*Carpenter et al., 2006*) to analyze our hN/hpA co-cultures after 21 d in vitro. We selected day 21 in vitro to study the developing synapse rather than a more mature synaptic network, which is not established in hNs at this timepoint. This decision was based in part on technical limitations associated with prolonged culturing of cells for screening applications, combined with the recognition that with current protocols, in vitro-derived neurons are unlikely to achieve a fully mature synaptic state regardless of time in culture. After immunostaining our co-cultures, image fields were acquired with a ×20 objective with the Opera Phenix high-content screening system (PerkinElmer), taking 12 fields per well and 5–8 stacks per field at 0.3 μm distance (*Figure 1—figure supplement 3*). We then used ALPAQAS pipelines to quantify MAP2-positive neurites and SYNAPSIN1 puncta parameters based on staining intensity and morphological thresholding (*Figure 1—figure supplement 3*). The first ALPAQAS pipeline was designed to merge in a single plane (maximum projection) the 5–8 stacks of each channel for each acquired field of a well. The generated output files were the inputs for a second pipeline, which corrected for potential illumination variations and translational misalignments, performed morphological reconstruction and de-clustering, and generated a *.csv file reporting at a field level: (1) the area covered by MAP2-positive neurites; (2) the number of SYNAPSIN1 puncta localized on MAP2-positive neurites; (3) the number of DAPI-positive nuclei; and (4) the area covered by SYNAPSIN1 puncta (*Figure 1—figure supplement 3*). To validate the specificity of synaptic detection in vitro, we assessed presynaptic density (i.e., the number of SYNAPSIN1 puncta localized on the MAP2-positive neurites divided by the area covered by MAP2-positive neurites) using ALPAQAS following small interfering RNA (siRNA) perturbation of SYNAPSIN1 (*SYN1*; Accell Dharmacon). Although siRNA-mediated knockdown is heterogeneous,

incubating human co-cultures with *SYN1* targeting siRNA for 72 hr significantly reduced presynaptic density compared to *GAPD* and non-targeting siRNA controls, with no or minimal impact on the area covered by MAP2-positive neurites or the total number of DAPI-positive neurites in any condition (*Figure 1g*; *Figure 1—figure supplement 4*). SYNAPSIN1 puncta with diameters between 1 and 6 pixels were selected based on analyses of their cumulative distribution (*Figure 1—figure supplement 4*), a setting which can be customized in our platform. Indeed, estimates of SYNAPSIN1 puncta diameter vary; some studies have used diameters of 0.4–1.6 µm$^2$ (*Yi et al., 2016*; *Patzke et al., 2019*), which would correspond to roughly 1–3 pixels in our assay (*Figure 1—figure supplement 4*).

Third, to convert the field-level analysis generated by ALPAQAS to well- or condition-level analyses, and to control for potential intra-plate edge and drift effects, we used Genedata Screener software (Genedata). Specifically, the *.csv data file generated from ALPAQAS was imported into Genedata through a high-content parser containing three additional quality control steps: (1) data were selected according to their MAP2 field area coverage in comparison to the intra-batch mean MAP2 field area coverage; (2) data from wells of poor quality (fewer than five fields persisting after the first quality control step) were discarded from the analysis; and (3) a minimum of two out of three wells per condition (e.g., genotype, treatment) was required by the parser for use in Genedata Screener to aggregate the data from a well-level to a condition-level (*Figure 1—figure supplement 5*). To control for potential intra-plate edge effects, we included two columns of control wells (0.1% DMSO treated) in each plate. In addition, two plates were assigned as sentinel or reference in each batch of human co-cultures to evaluate and correct eventual anomalies due to intra-batch drift effects. We then leveraged pattern correction algorithms in Genedata Screener to potentially rectify values across plates prior to performing Z-score calculations (*Sherman and Bang, 2018*; *van der Kant et al., 2019*; *Zhang et al., 1999*) for synaptic density (*Figure 1c*; *Figure 1—figure supplement 5*).

Collectively, these customized protocols and analysis pipelines establish a novel platform for automated quantification of presynaptic density of hN + hpA co-cultures.

## Primary screening results for 376 small molecules followed by secondary validation reveal six potent small molecules increasing presynaptic density

To validate the robustness of our platform and uncover modulators of human synaptogenesis, we next assessed the impact of 376 small molecules from a highly selective inhibitor library (SelleckChem, L3500, *Supplementary file 1*). Here, small molecules were not selected based on known roles in neuronal or synaptic development, but instead from structurally diverse classes of inhibitors targeting kinases, chromatin modifiers, and cytoskeletal signaling pathways, among others (*Supplementary file 1*, *Figure 2—figure supplement 1*). Human co-cultures (hNs + hpAs) were treated with small molecules at a concentration of 3 µM in triplicate for 72 hr starting on day 18 of hN differentiation (*Figure 2a*). Importantly, 92.94% of the control and treated wells had a Z-score for synaptic density between –2 and +2, underscoring the reliability and robustness of our automated high-content synaptic screening platform (*Figure 2—figure supplement 1*). Thirteen small molecules (3.46% of the library) did not meet the above quality control requirements and seven (1.86% of the library) reduced the density of SYNAPSIN1 puncta on MAP2 neurites by a Z-score ≤–3 and reduced the area covered by MAP2-positive neurites by a Z-score ≤ –2 (*Figure 2a* and *Figure 2—figure supplement 1*); as expected, these small molecules were known to induce apoptosis or catalytic process of autophagy (*Boccadoro et al., 2005*; *Piva et al., 2008*; *Reddy et al., 2011*). Representative examples of the effects of individual small molecules on presynaptic density are shown in *Figure 2b*.

To further validate the top hit classes that increased presynaptic density in our primary screen, we executed secondary dose–response assays with newly purchased reagents. Specifically, we selected three heat-shock protein 90 inhibitors (Luminespib, Ganestespib, and Tanespimycin), three BET inhibitors (I-BET151 and (+)-JQ1 included in the primary screen in addition to Birabresib, which was added to expand the number of independent BET inhibitors queried), two phosphoinositide 3-kinase inhibitors (VS-5584 and TG100-15), two histone deacetylase inhibitors (Tubacin and Belinostat), two HMG CoA reductase inhibitors (Fluvastatin and Lovastatin), an S6 ribosome inhibitor and a ROCK1/ROCK2 inhibitor (BI-D1870 and GSK429286A, respectively), a histamine H1 and an opioid receptor antagonist (Loratadine and Naltrexone, respectively), and a ligand of ubiquitin E3 (Lenalidomide) (*Figure 2a and b*), for a total of 17 small molecules. We then performed dose–response assays in hN + hpA co-cultures

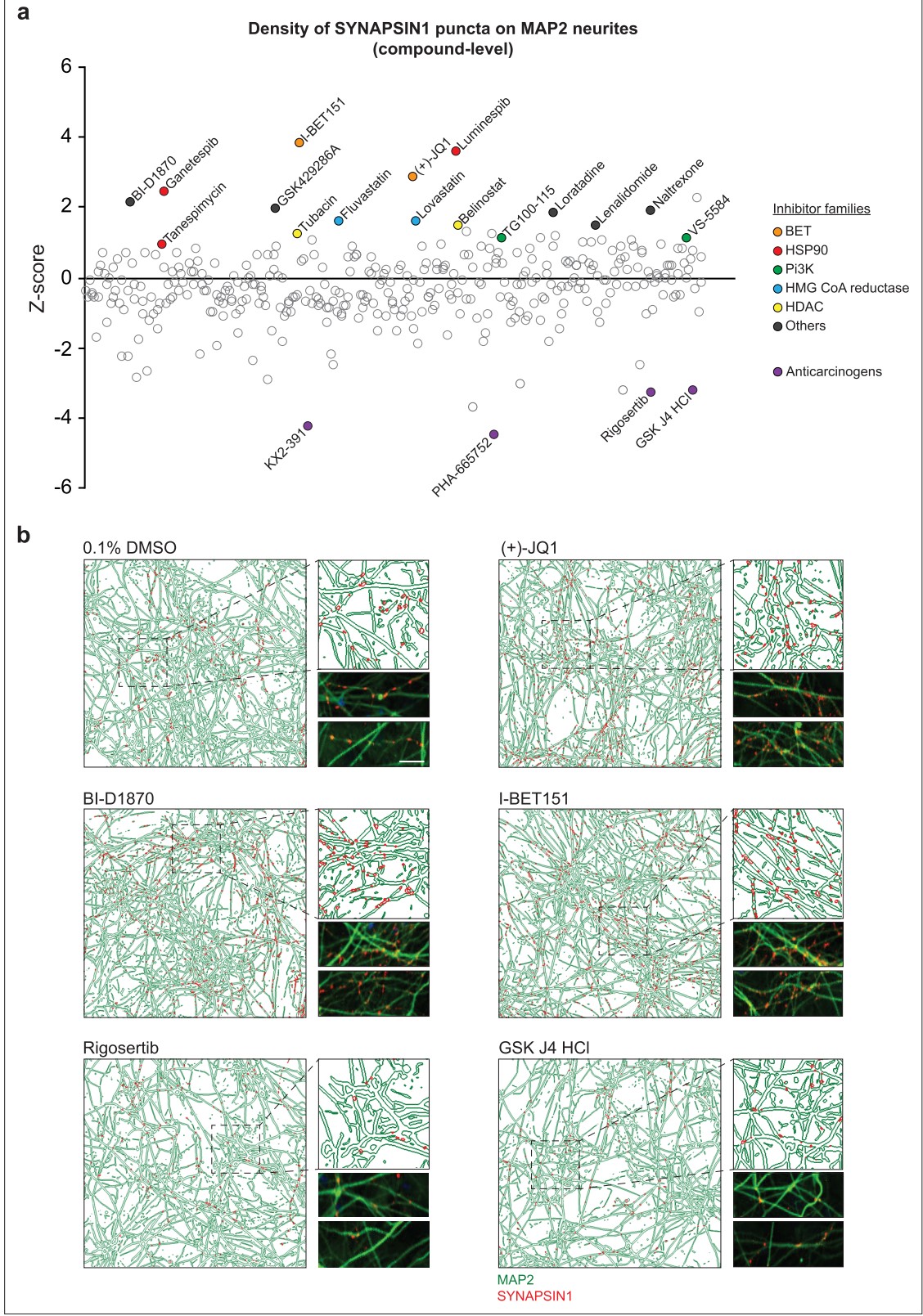

**Figure 2.** Primary screening results for 376 small molecules. (**a**) Z-score values for the density of SYNAPSIN1 on MAP2-expressing neurites for hN + hpA co-cultures incubated with small molecules in triplicate at 3 µM for 72 hr starting on day 18 of neuronal differentiation (each circle represents the small molecule level aggregate of the replicates). The effects of each individual small molecule were assessed using our in silico pipelines. Small molecules that increased presynaptic density and were selected for further validation are indicated by colored circles. Those in purple are anticarcinogens that

*Figure 2 continued on next page*

*Figure 2 continued*

reduced presynaptic density and were not selected for further validation. (**b**) Representative immunofluorescence images and CellProfiler output images (field-level) of human co-cultures treated with either 0.1% DMSO, (+)-JQ1, BI-D1870, I-BET151, Rigosertib, or GSK J4 HCl. Scale bar = 1 µm. Note the increase in SYNAPSIN1 puncta on MAP2-expressing neurites following (+)-JQ1, BI-D1870, and I-BET151 compared to 0.1% DMSO control, and the decrease in SYNPASIN1 puncta following treatment with Rigosertib and GSK J4 HCl.

The online version of this article includes the following figure supplement(s) for figure 2:

**Figure supplement 1.** Small molecules used in the primary screen.

using the automated HP D300e Digital Dispenser (HP, FOL57A) to reliably and randomly dispense at increasing concentrations, nanoliters or microliters of the freshly dissolved small molecules across the 60 inner wells of each 96-well plate, and subsequently quantified synaptic density using our previously described ALPAQAS pipelines (*Figure 3a*).

We found that 6 out of 17 small molecules tested exhibited a significant dose-dependent increase in presynaptic density compared to DMSO-treated controls (*Figure 3* and *Figure 3—figure supplements 1 and 2*). Notably, all three BET inhibitors ((+)-JQ1, Birabresib, and I-BET151) increased presynaptic density with similar magnitudes of effect (*Figure 3b and c* and *Figure 3—figure supplements 1 and 2*). BI-D1870 and GSK429286A also elicited a significant dose–response, indicating that the synaptic connectivity in hNs is modulated through multiple intracellular pathways (*Figure 3—figure supplements 1 and 2*). Indeed, previous studies have shown that ROCK1/ROCK2 inhibition can enhance synapse formation in rodent models (*Swanger et al., 2015*), consistent with our results from GSK429286A treatment of hN + hpA co-cultures. Finally, VS-5584, an ATP-competitive Pi3K inhibitor with equivalent potency against all human isoforms (α, β, γ, and δ), drove a concentration-dependent increase in presynaptic density compared to the DMSO-treated wells, but not TG100-15, a selective inhibitor of the γ and δ isoforms (*Figure 3d* and *Figure 3—figure supplement 1*), consistent with the preferential immune-cell expression of Pi3Kγ and δ versus the ubiquitous expression patterns of Pi3Kα and β (*Winkler et al., 2013*). The six validated small molecules showed minimal to no toxicity or effect on neurite outgrowth at lower doses, as assessed by the number of DAPI-positive nuclei detected and the area covered by MAP2-positive neurites (*Figure 3b–d* and *Figure 3—figure supplements 3 and 4*).

For subsequent experiments, we determined the optimal dosage of each small molecule in hN + hpA co-cultures based upon (1) a significant increase in synaptic connectivity compared to the DMSO-treated wells; (2) a concentration higher than the EC (50); (3) minimal impact on the area covered by MAP2-positive neurites; and (4) minimal or no toxicity (*Figure 3* and *Figure 3—figure supplements 1–4*). We selected 0.5 µM for (+)-JQ1, 1 µM for Birabresib, 3 µM for I-BET151, VS-5584, and BI-D1870, and 5 µM for GSK429286A as effective concentrations to increase human presynaptic density in hN + hpA co-cultures. We then derived hNs from an independent hPSC line with the same stable, inducible AAVS1 NGN2 integration strategy described above, and confirmed that four out of six small molecules, including all three BET inhibitors, increased presynaptic density in hN + hpA co-cultures at these specific concentrations (*Figure 3—figure supplement 5*), further validating our previous findings and indicating that the effects of the selected small molecules were generally reproducible across cell lines. Collectively, these analyses highlight the robustness and the sensitivity of our platform to detect modulators of human synaptogenesis.

## Astrocytes are key regulators of human synapse assembly in vitro and response to small molecules

Astrocytes are known to play critical roles in the regulation of neuronal network development (*Chung et al., 2015*; *Perez-Catalan et al., 2021*), including the enhancement of presynaptic function (*Ullian et al., 2001*), and we therefore assessed the necessity of hpAs in mediating the effects of the identified small molecules. This was particularly relevant to assess, given that astrocytes are not standardly included in hN screening strategies. Specifically, we tested the impact of the six validated small molecules described above on presynaptic density in the absence of hpAs. hN monocultures were plated, treated, and analyzed using our established pipelines (*Figure 4a*). While hN monocultures passed all quality control and thresholding criteria (*Figure 4—figure supplement 1*), none of the six small molecules elicited a significant increase in presynaptic density in this condition (*Figure 4b*).

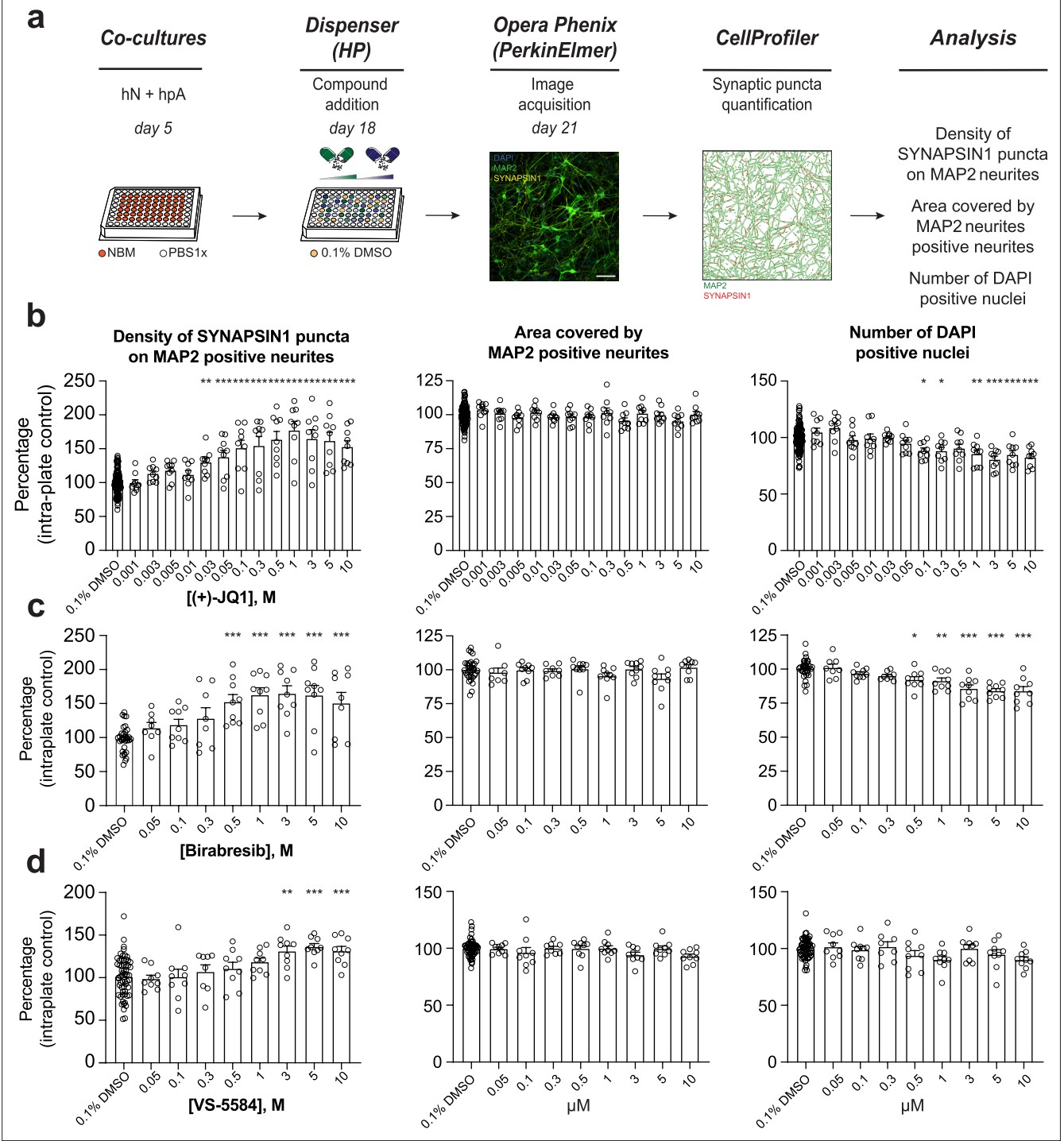

**Figure 3.** Secondary validation reveals potent small molecules increasing presynaptic density. (**a**) Workflow for the dose–response assay. hN + hpA co-cultures were treated on day 18 with selected small molecules at various concentrations or 0.1% DMSO for 72 hr, stained and processed through the synaptic assay on day 21. Scale bar = 100 pixel. (**b–d**) Concentration responses for three small molecules (two BET inhibitors [(+)-JQ1, Birabresib] and VS-5584) increasing human presynaptic density (left). The impact on the area covered by MAP2-positive neurites (middle) and toxicity (number of DAPI-positive nuclei; right) are also shown. Data are quantified by percentage of intra-plate control (0.1% DMSO) represented as mean values ± SEM, n = 3 biological replicates, n = 3 technical replicates. *p<0.05, **p<0.01, ***p<0.001; one-way ANOVA with Dunnett's multiple comparisons test.

The online version of this article includes the following figure supplement(s) for figure 3:

**Figure supplement 1.** Dose–response curves for the impact of 17 selected small molecules on human presynaptic density.

**Figure supplement 2.** Histograms for the impact of 17 selected small molecules on human presynaptic density.

*Figure 3 continued on next page*

*Figure 3 continued*

**Figure supplement 3.** Histograms for the impact of 17 selected small molecules on the area covered by MAP2.

**Figure supplement 4.** Histograms for the impact of 17 selected small molecules on the number of DAPI-positive nuclei.

**Figure supplement 5.** The effect of BET inhibition on presynaptic density is reproducible across cell lines.

As expected, the area covered by MAP2-positive neurites and the number of DAPI-positive nuclei remained unchanged by small molecule treatment (*Figure 4c and d*), paralleling the hN + hpA co-culture condition (*Figure 3*). Importantly, we performed experiments with and without hpAs in parallel using experiments with hpAs as a positive control for experiments performed without hpAs. Small molecule treatment also did not significantly affect the level of SYNAPSIN1 protein expression in either the hN monoculture or hN + hpA co-culture conditions as assessed by western blot analysis (*Figure 4e and f*), indicating that the differences observed between hN monocultures and hN + hpA co-cultures after small molecule addition were more likely due to changes in SYNAPSIN1 protein localization as opposed to overall protein abundance.

The addition of hpAs also had a significant effect on synapse development in the absence of small molecules, including a significant increase in SYNAPSIN1 presynaptic density as well as in the size of individual SYNAPSIN1 puncta with a modest reduction in the area covered by MAP2-positive neurites (*Figure 5a–e*). Indeed, the addition of hpAs roughly doubled both the density and size of SYNAPSIN1 puncta on MAP2-positive neurites compared with hNs alone (*Figure 5c and d*). Collectively, our results indicate that hNs can form quantifiable presynaptic puncta that can be reliably captured by our platform in the absence of astrocytes. However, hpAs contributed to the assembly or localization of presynaptic machinery, underscoring the relevance of astrocytes in human synaptic screening applications. hpAs were therefore included in all subsequent experiments.

## Power calculations inform on study design

To more precisely define the power of our assay and gain additional insight into study design, we calculated effect size (Cohen's d) for our primary screen overall as well as for each small molecule tested, for our hN + hpA co-culture validation experiments, and for our hN monoculture validation experiments (*Figure 5f and g*, *Figure 5—figure supplement 1*, *Supplementary file 2*). For the primary screen, we obtained a Cohen's d of –0.18, as individual small molecules increased as well as decreased presynaptic density (*Figure 5—figure supplement 1*, *Supplementary file 2*). Data for each small molecule included in the primary screen is shown in *Supplementary file 2*. Notably, both (+)-JQ1 and I-BET151 had very large effect sizes in the primary screen, with a Cohen's d of 2.914 for (+)-JQ1 and 3.710 for I-BET151 (*Supplementary file 2*). Consistent with the results of the primary screen, we also noted large effect sizes for each BETi tested in our hN + hpA co-culture validation experiments (*Figure 5f and g*, *Figure 5—figure supplement 1*, *Supplementary file 2*). Here, effect sizes were sufficiently large that the number of replicate fields (n = 47–80) and wells (n = 6–9) analyzed per small molecule could be scaled down in future experiments (*Figure 5f and g*, *Figure 5—figure supplement 1*, *Supplementary file 2*). For our hN monoculture validation experiments, we noted effect sizes in the small to medium range (*Figure 5f and g*, *Figure 5—figure supplement 1*, *Supplementary file 2*). Here, we scaled up the number of fields (n = 404–417) and wells (n = 52) analyzed per small molecule, which was sufficient to achieve 95% power for detecting the small and medium effect sizes of Birabresib and I-BET151 (one-sided power = greater; *Figure 5f and g*, *Figure 5—figure supplement 1*, *Supplementary file 2*). These analyses confirmed that we were well powered to detect the impacts of BETi in both the hN co-culture and monoculture assays; however, the hN monoculture condition required substantially greater scale given the reduced effect sizes. Overall, in order to achieve 95% power (one-sided) to detect small effect sizes (Cohen's d = 0.2), experiments would require around n = 270 replicates, while medium effect sizes (Cohen's d = 0.5) would require around n = 43 replicates, and large effect sizes (Cohen's d = 0.8) would require around n = 17 replicates. In future experiments, small molecules with effect sizes similar to those of the BET inhibitors in the co-culture condition could be readily detected with modest experimental scale as noted above; in order to detect small effect sizes, experiments may require hundreds of replicates.

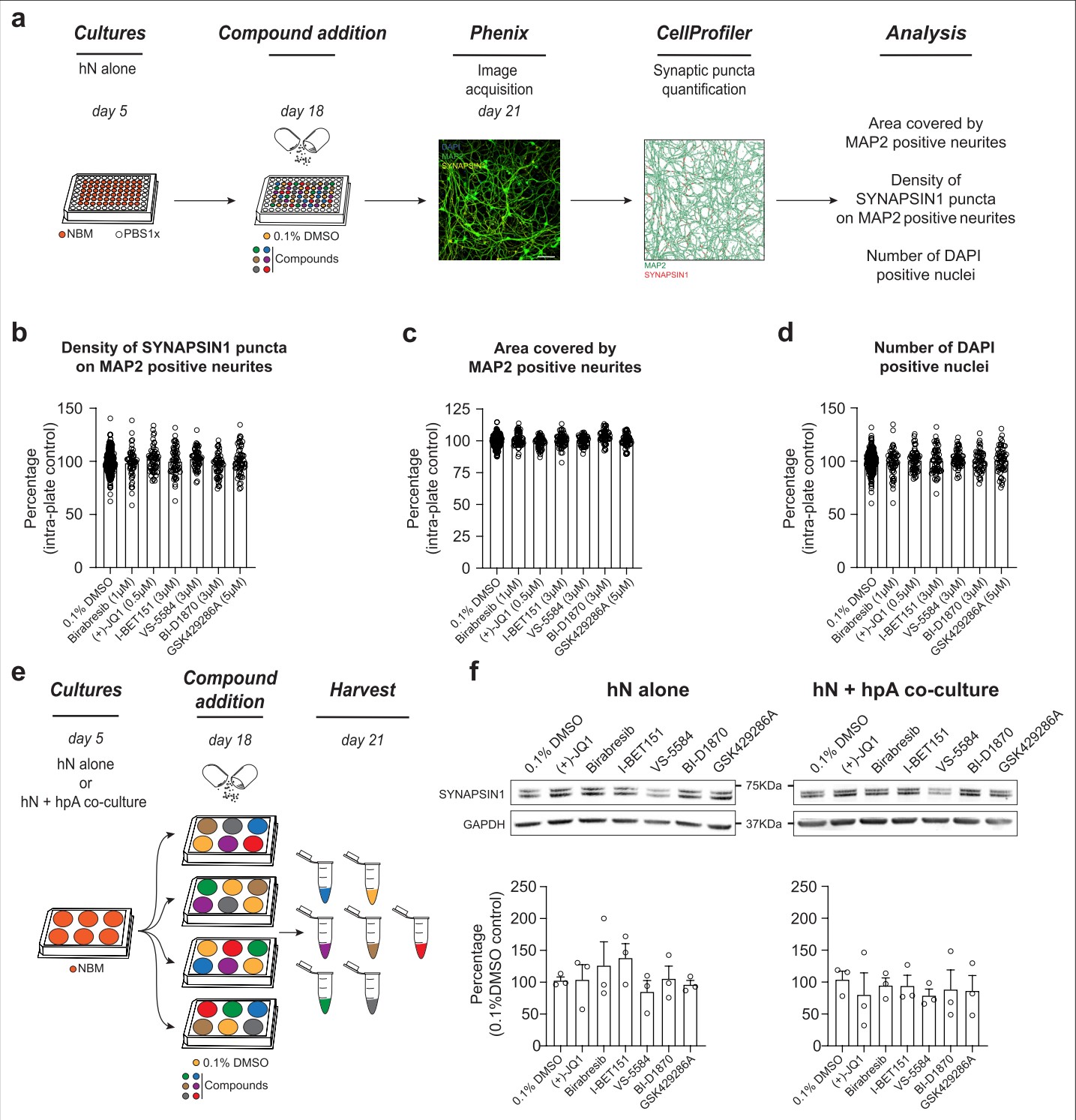

**Figure 4.** Astrocytes play a critical role in mediating response to small molecules. (**a**) Workflow for small molecule validation in the absence of hpAs. Scale bar = 100 pixel. (**b–d**) In the absence of hpAs, none of the selected small molecules affected SYNAPSIN1 density of hNs, in contrast to the co-culture condition analyzed in parallel. The area covered by MAP2-positive neurites and cell viability was also not impacted as compared to intra-plate DMSO controls. Error bars are shown as mean +/- SEM. (**e**) Schematic of the preparation of the immunoblot samples. (**f**) Representative images of immunoblots and quantification of SYNAPSIN1 normalized to GAPDH and presented as a percentage of DMSO control in both hN alone and hN + hpA co-culture conditions. Note that SYNAPSIN1 protein expression levels were not impacted in any condition. n = 3 biological replicates (**f**). Error bars are shown as mean +/- SEM.

*Figure 4 continued on next page*

*Figure 4 continued*

The online version of this article includes the following source data and figure supplement(s) for figure 4:

**Source data 1.** Source of western blots for SYNAPSIN1 as well as GAPDH loading control following small molecule treatment versus DMSO control.

**Figure supplement 1.** Acceptance criteria of all batches.

## Multiple BET inhibitors enhance synaptic gene expression

Given that BET inhibitors were the most prominent hit class identified in our small molecule screen, with three independent BET inhibitors increasing presynaptic density in two independent hPSC lines, we sought to confirm a role for BET proteins in human synaptic development and further validate the results of our synaptic assay. The BET family of chromatin readers, which includes BRD2, BRD3, BRD4, and BRDT, bind acetylated lysine residues on histone proteins as well as transcription factors to mediate gene expression. Through unbiased screening, (+)-JQ1 was identified as a positive modulator of human neurogenesis (*Li et al., 2016*). However, in adult mice, BET inhibition through (+)-JQ1 was shown to reduce synaptic gene expression and impair memory consolidation (*Korb et al., 2015*), while BET inhibition through I-BET858 in mouse primary neurons reportedly led to decreased expression of neuronal differentiation and synaptic genes (*Sullivan et al., 2015*). Importantly, region-specific differences in response to (+)-JQ1 have been identified in the rodent brain, particularly in the context of dendritic spine density, suggesting that brain cell type may be a major determining factor in the expression of these phenotypes (*Wang et al., 2021b*). Moreover, a recent study found sex divergent effects of Brd4 on cellular and transcriptional phenotypes in both human and mouse (*Kfoury et al., 2021*), further supporting the hypothesis that the impacts of BET inhibition are highly context dependent.

We therefore performed global transcriptional analyses on both (+)-JQ1 and Birabresib-treated hN + hpA co-cultures compared with DMSO-treated controls (*Figure 6a*) to independently assess their roles in synaptic development. (+)-JQ1, Birabresib, or DMSO was added on day 18 in vitro for 72 hr, paralleling the approach used in our small molecule screen. Applying an adjusted p-value cut-off of 0.05 and $\log_2$ fold change cut-offs of $\leq -1$ and $\geq 1$, we identified a comparable number of significantly differentially expressed genes (DEGs) after (+)-JQ1 and Birabresib treatment compared to DMSO control (*Figure 6b and c*, *Supplementary files 3 and 4*). A majority of DEGs were downregulated after BET inhibition (*Figure 6b and c*), consistent with the known roles of BET proteins in transcriptional activation, and were also shared between the (+)-JQ1 and Birabresib treatment conditions (n = 2368; *Figure 6d*). We also observed strong positive correlation in the magnitudes of effect of (+)-JQ1 and Birabresib treatment as attested by the Pearson's r score of 0.9404 (*Figure 6e*). Given that both BET inhibitors increased presynaptic density (*Figure 3b and c*), we focused on the set of DEGs that were shared between (+)-JQ1 and Birabresib treatment conditions for the remainder of our transcriptional analyses. Using ingenuity pathway analysis (IPA), we confirmed that BRD4 (p=$1.61 \times 10^{-17}$), (+)-JQ1 (p=0.000206), and Birabresib (p=$4.59 \times 10^{-18}$) were all strongly predicted to be upstream regulators of the shared DEGs (*Figure 6—figure supplement 1*). Signaling pathways such as 'DNA methylation and transcriptional repression' and 'axonal guidance' were enriched in the upregulated DEGs, while signaling pathways such as 'wound healing' and 'neuroinflammation' enriched in the downregulated DEGs (*Figure 6f*). We then leveraged SynGO (*Koopmans et al., 2019*) to specifically analyze synaptic ontology terms, focusing on genes upregulated in both the (+)- JQ1 and Birabresib transcriptional datasets (*Figure 6g and h*, *Supplementary file 5*). We found significant enrichment for the terms 'integral component of postsynaptic membrane' and 'postsynaptic membrane' with modest but significant enrichment for additional presynaptic terms (*Figure 6g and h*, *Supplementary file 5*) consistent with a role for BET proteins in the regulation of pre- and postsynaptic genes. Examples of individual pre- and postsynaptic genes upregulated following BET inhibition include the presynaptic cell adhesion molecule *Neurexin 3 (NRXN3)*, the synaptic vesicle gene *Amphiphysin (AMPH)*, the transmembrane Eph receptor ligand *Ephrin-B2 (EFNB2)* involved in AMPAR stabilization, and the postsynaptic scaffolding factors *BAR/IMD Domain Containing Adaptor Protein 2 (BAIAP2)* and *Homer Scaffold Protein 1 (HOMER1)* (*Figure 6i*). Indeed, (+)-JQ1 and Birabresib treatment each roughly doubled the expression of these pre- and postsynaptic genes. Additionally, western blot analysis revealed that both (+)-JQ1 and Birabresib significantly increased protein expression levels

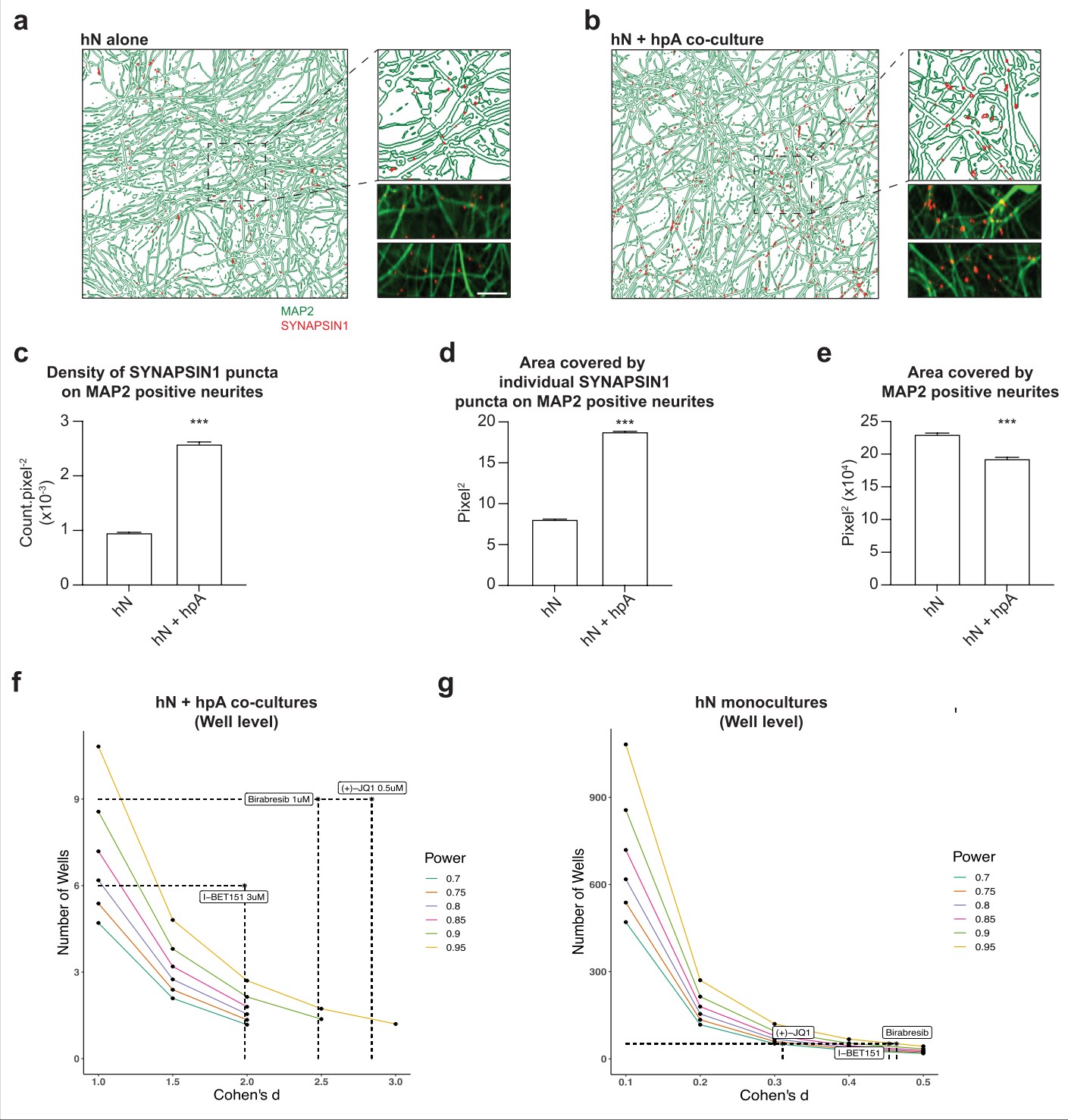

**Figure 5.** Astrocytes are key regulators of human synapse assembly in vitro. (**a, b**) Representative immunofluorescence and CellProfiler output images (field-level) of hN monocultures (**a**) and hN + hpA co-cultures (**b**). Scale bar = 1 μm. (**c–e**) Measurements comparing the density of SYNAPSIN1 puncta on MAP2-positive neurites (**c**), the area covered by individual SYNAPSIN1 puncta (**d**) and the area covered by MAP2-positive neurites (**e**) between hN monocultures and hN + hPA co-cultures. Data are represented as mean values ± SEM, n = 3 biological replicates, n = 8 technical replicates (**c–e**); n > 1000 fields and n > 50,000 SYNAPSIN1 puncta for each condition (**c–e**). ***p<0.001; Kolmogorov–Smirnov unpaired *t*-test. (**f, g**) Power calculations for hN + hPA co-culture (**f**) and hN monoculture (**g**) validation experiments at the well level for the hypothesis 'greater than.' Cohen's d and the number of wells analyzed for each experiment following 0.5 uM (+)-JQ1, 3 uM I-BET151, and 1 uM Birabresib treatment are indicated by the dotted lines.

The online version of this article includes the following figure supplement(s) for figure 5:

**Figure supplement 1.** Power calculations for the primary screen and validation experiments.

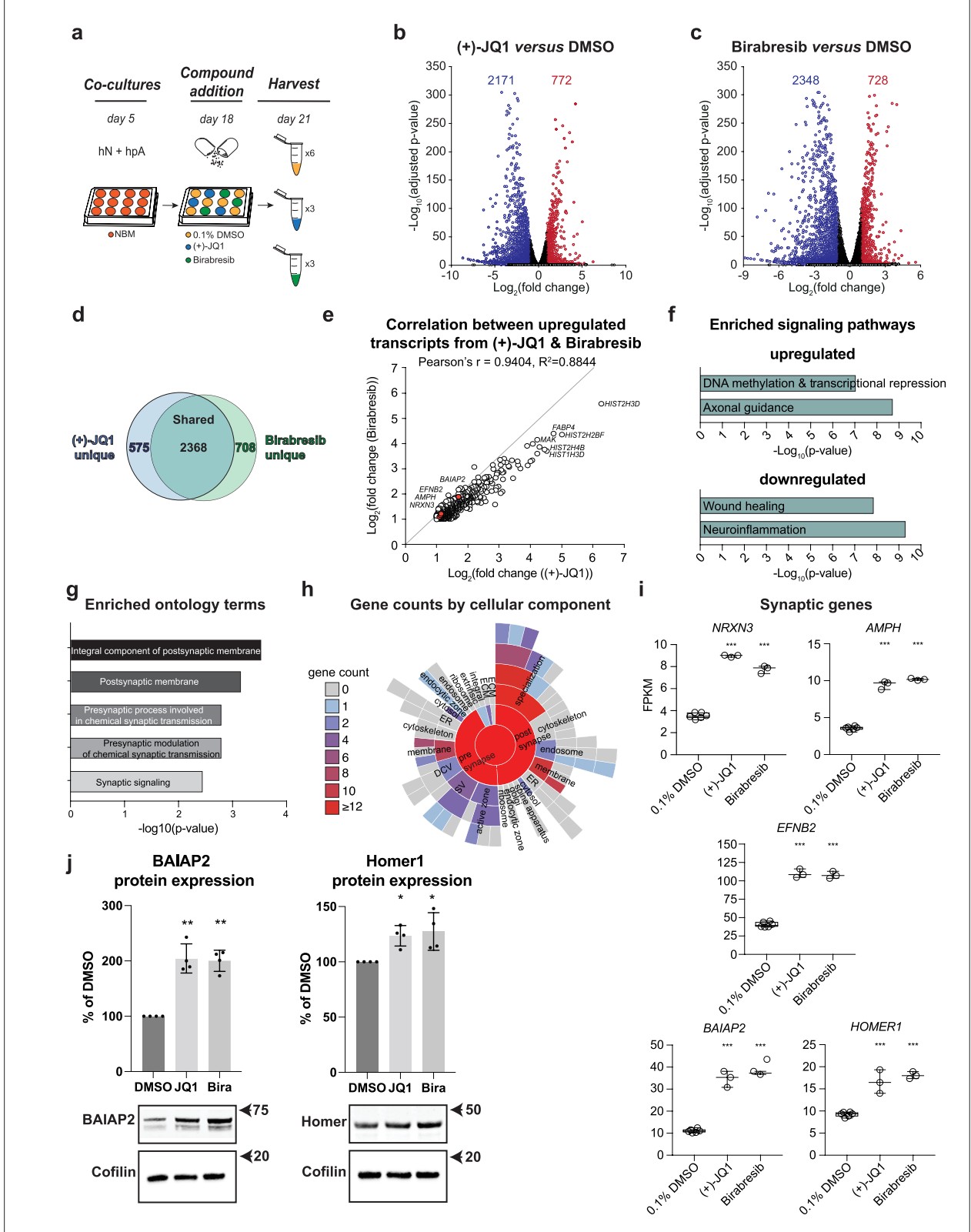

**Figure 6.** Multiple BET inhibitors enhance synaptic gene expression. (**a**) Schematic of mRNA-seq experiment using 3–6 replicates per treatment condition. hN + hpA co-cultures were treated with 0.1% DMSO, 0.5 μM (+)-JQ1 or 1 μM Birabresib for 72 hr prior to harvesting. (**b, c**) Volcano plots for (+)-JQ1 versus DMSO control (**b**) and Birabresib versus DMSO control (**c**). Log₂ fold change is shown on the x-axis, with the -log₁₀ of the adjusted p-value shown on the y-axis. Positive fold change reflects an increase in the treatment condition relative to DMSO control. Numbers of significantly differentially

*Figure 6 continued on next page*

*Figure 6 continued*

expressed genes (DEGs) are shown for each condition. (**d**) Overlap (shown as number of genes) between DEGs identified with (+)-JQ1 treatment (blue) and DEGs identified with Birabresib treatment (green). (**e**) Scatterplot of the fold change read counts of the genes upregulated by (+)-JQ1 and Birabresib. Note the strong positive Pearson's r (94%) correlation between transcripts upregulated by (+)-JQ1 and Birabresib. Select gene examples are highlighted in red. (**f**) Select canonical pathways identified by IPA for DEGs upregulated in both (+)-JQ1 and Birabresib treatment conditions (top) or downregulated in both (+)-JQ1 and Birabresib treatment conditions (bottom). The x-axis shows -$\log_{10}$ of the p-value for each pathway analysis term. (**g**) SynGO analysis showing the top 5 significantly enriched cellular component and biological process ontology terms identified using all genes upregulated by (+)-JQ1 and Birabresib. The x-axis shows -$\log_{10}$ of the p-value for each ontology term; the top two terms also met a<1% FDR threshold (**h**) SynGO analysis sunburst representation showing gene counts of the cellular components identified using all genes upregulated by (+)-JQ1 and Birabresib. (**i**) Graphs of transcript expression values for pre- and post-synaptic genes. Significance was calculated by Benjamini–Hochberg adjusted Wald test as part of the DEseq2 RNA-seq experiment. n=3 biological replicates. (**j**) Western blot analysis of BAIAP2 and Homer1 protein expression levels following treatment with DMSO, (+)-JQ1, or Birabresib (n = 4 biological replicates). Top, quantification (shown as % of DMSO control); bottom, example blots. For all figure panels, significance is indicated by *p<0.05, **p<0.01, ***p<0.001 relative to controls.

The online version of this article includes the following source data and figure supplement(s) for figure 6:

**Source data 1.** Source of western blots for BAIAP2 and Homer1 as well as Cofilin loading control following DMSO, (+)-JQ1, or Birabresib treatment conditions.

**Figure supplement 1.** BET proteins and inhibitors are identified as upstream regulators of the differentially expressed genes (DEGs) and BET inhibition impact astrocyte-secreted and pro-inflammatory factors.

of the postsynaptic scaffolding factors BAIAP2 and Homer1 compared with DMSO-treated control (*Figure 6j*), paralleling the transcript-level changes.

Consistent with the pathway analyses, we also observed modulation of the expression of several astrocyte-secreted factors including upregulation of the anti-inflammatory factor *Interleukin 11 (IL11)*, downregulation of the pro-inflammatory factor Interleukin 6 (*IL6*), and downregulation of the synapse disassembly factor *Secreted Protein Acidic And Cysteine Rich (SPARC)* (*Figure 6—figure supplement 1*). Combined with downregulation of Caspase 1 (*CASP1)* and *Gasdermin D (GSDMD)* (*Figure 6—figure supplement 1*), these results point to a general anti-inflammatory response following BET inhibition. Indeed, *CASP1, GSDMD,* and *IL6* have all previously been shown to be downregulated by (+)-JQ1 treatment, consistent with the known role of BET proteins in promotion of the inflammatory response (*Wang et al., 2021b*; *Zhou et al., 2019*). Importantly, changes in gene transcription were again highly concordant between both (+)-JQ1 and Birabresib treatment conditions, with similar magnitudes of effect (*Figure 6—figure supplement 1*).

Collectively, these data indicate that BET inhibition can lead to upregulation of synaptic genes, consistent with our screening results showing an increase in presynaptic density and supporting a role for BET proteins in synapse assembly. These results also support the robustness of our platform for detecting human synaptic modulators in vitro.

## Discussion

Despite overwhelming evidence implicating synaptic dysfunction in complex human disease and clear species-specific differences in certain aspects of synaptic biology, human neurons and astrocytes have yet to be widely employed in studies of synaptic mechanisms. We therefore established a novel, automated, high-content synaptic platform using human neurons and astrocytes to facilitate dissection of human synaptic mechanisms and drug discovery efforts. Specifically, we (1) generated stable iNGN2-hPSC lines through TALEN editing to produce large, reproducible batches of postmitotic glutamatergic hNs; (2) optimized hN + hpA co-culture and immunocytochemistry conditions for reliable quantification of synaptic markers on an automated high-content imaging platform; (3) employed liquid handling systems to standardize the dispensing steps and maximize accuracy for both co-culturing and immunocytochemistry; and (4) developed in silico automated pipelines for rigorous assessment of plating consistency and quantification of human synaptic connectivity. After extensive vetting of antibodies for synapse detection, SYNAPSIN1 on MAP2 was the only antibody combination sufficiently robust to be captured in our assay. While there is no single definitive synaptic marker, SYNAPSIN1 is one of the most robust markers utilized in the field, having been shown to label the vast majority of rodent glutamatergic cortical synapses (*Micheva et al., 2010*) and frequently employed in studies of human cellular systems (*Chanda et al., 2019*; *Pak et al., 2015*; *Yi et al., 2016*).

The incomplete co-localization of SYNAPSIN1 with synaptophysin and PSD-95 could reflect factors such as developmental stage, as SYNAPSIN1 expression has been reported to precede synaptophysin and PSD-95 expression (*Friedman et al., 2000*; *Pinto et al., 2013*), or the weaker signal obtained by immunocytochemistry for the latter antibodies. Development of additional synaptic antibodies suited for immunocytochemical applications in human cells will be essential to more definitively address this question.

Of note, several open-source methods for synaptic quantification have been developed in recent years including SynQuant, SynapseJ, SynPAnal, and SynD (*Danielson and Lee, 2014*; *Moreno Manrique et al., 2021*; *Schmitz et al., 2011*; *Wang et al., 2020*) providing a solid foundation for the principle of synaptic quantification from immunofluorescence images. However, our platform is the first to use human neurons and human astrocytes for scaled analyses of synapses, and one of the few to fully automate each step of the process from co-culture to immunostaining to imaging, in order to enhance throughput, accuracy, and reproducibility. Through unbiased screening of 376 small molecules, we generated proof-of-concept data that our platform works robustly to identify synaptic modulators. Our results were also generally reproducible across independent hPSC lines. Given the known molecular and functional heterogeneity across individual hPSC lines (*Kajiwara et al., 2012*; *Kilpinen et al., 2017*; *HIPSCI Consortium et al., 2018*), we would not expect identical results or magnitudes of effect, but our analyses support the robustness of our assay to detect differences in independent hPSC lines.

Importantly, the results from hNs co-cultured with hpAs did not correlate with the results from hNs cultured alone, underscoring the importance of including astrocytes in human synaptic assays. Indeed, our positive hits required the presence of hpAs to achieve a significant change in presynaptic density, supporting the relevance of non-cell-autonomous factors in synapse assembly. Our power calculations comparing the hN monoculture condition with the hN + hpA co-culture condition suggest we are reasonably powered to detect changes in both assays given the predicted effect sizes. While we cannot rule out the possibility of technical factors contributing to the lack of changes in presynaptic density in the hN monocultures, as opposed to differences in the biological contribution of hpAs, it is critical to note for future screening applications that the inclusion of astrocytes was either required for these biological changes or important for their practical detection. Indeed, large effect changes seen in the co-culture condition could be detected with a small number of replicates (e.g., three wells), while the monoculture condition required 50+wells for the same small molecules. Consistent with astrocytes playing a critical role in synapse formation, upon hpA co-culture, we observed a twofold increase in presynaptic density and the area of individual SYNAPSIN1 puncta compared with hN monocultures. While previous studies have shown that inclusion of rodent astrocytes with hNs can significantly increase physiological and/or morphological maturation (*Johnson et al., 2007*; *Nehme et al., 2018*; *Tang et al., 2013*), the underlying mechanisms remain incompletely resolved. Studies examining the effects of physical astrocyte contact versus astrocyte conditioned media on hNs report that physical contact has a much more potent impact on synapse development compared with conditioned media (*Johnson et al., 2007*; *Tang et al., 2013*), suggesting that direct physical contact between hNs and hpAs may be particularly important. Our results imply that regulators of synapse formation may be missed in assays performed in the absence of astrocytes. Given that the inclusion of astrocytes more accurately mimics the in vivo synaptic milieu, and that astrocytes are not standardly included in hN screening strategies, it will be critical to consider this cell type in future human synaptic screening experiments.

Given that three separate BET inhibitors scored in our assay in two independent hPSC lines, our results also support a role for BET proteins in human synaptic development. While we were unable to separate hN and hpA transcriptomes from the co-culture (and the co-culture was required to see the effects of BET inhibition), we observed modulation of synaptic machinery in our global transcriptional analyses, consistent with our screening results. These data support a model whereby BET inhibition either leads to increased expression of specific synapse associated genes, or to the enhancement of presynaptic puncta which then triggers the expression of additional synaptic components. Indeed, BET protein expression is downregulated throughout neuronal differentiation, with knockdown of BET proteins sufficient to enhance differentiation (*Li et al., 2016*) consistent with a role for BET proteins in negatively regulating neuronal development. These results contrast with a subset of data generated in adult mouse brain and are thus likely to be dependent upon factors such as brain cell type,

developmental stage, and timing of drug treatment. Resolving potential neurological benefits and/or consequences of BET inhibition in different contexts including in diverse brain cell types and developmental stages is essential as these drugs are currently in use in clinical trials for cancer and are proposed for the treatment of developmental as well as degenerative diseases (*Korb et al., 2015*; *Wang et al., 2021a*).

While BET proteins have well-described roles in the inflammatory response, little is known about their role in astrocytes specifically. Future studies will be required to determine how BET inhibition impacts astrocytes and how this may in turn impact neuron function. Our results are consistent with BET inhibition enhancing synapse assembly and impacting both neuron and astrocyte transcriptomes, but it remains to be determined whether BET inhibition of astrocytes has a direct impact on synapse formation or whether BET inhibition primarily acts on neurons to impact synapse assembly. Overall, we speculate that hpAs provide critical synaptic support for hNs in our assay that then facilitates the health or receptiveness of hNs for further synaptic enhancement, given the disparities in SYNAPSIN1 parameters between hN monocultures and hN + hpA co-cultures, as well as the lack of effect of small molecules on hN monocultures.

Our robust, scalable and accessible platform provides several key advances for the field. Specifically: (1) we generated the first automated and quantitative high-content synaptic phenotyping platform for human neurons and astrocytes, which can now be leveraged for study of basic human synaptic mechanisms, dissection of complex human disease and drug discovery efforts; (2) our results demonstrate that the stable NGN2 integration strategy for scaled generation of hNs from hPSCs is well suited for screening applications; (3) our platform is compatible with both small molecule as well as siRNA-based manipulations; (4) our platform captures key parameters of human synaptic connectivity including the number and the area of individual presynaptic contacts as well as the coverage of MAP2-positive neurites and overall cell viability; (5) while compatible with analyses of hN monocultures, our analyses underscore the relevance of astrocytes when studying human synaptic mechanisms and drug response and (6) different human genetic backgrounds can be employed in our platform to dissect disease mechanisms. Future studies will focus on the incorporation of additional brain cell types to increase the complexity in our 2D co-culture systems, probing additional small molecule classes to explore novel human synaptogenic factors and improving the fidelity of astrocyte differentiation protocols in vitro to facilitate cell-type-specific genetic manipulations.

There are also several limitations to our approach: (1) our platform was specifically designed for use with human neurons and astrocytes and alterations to the neuronal differentiation paradigms or species, or the addition of other brain cell types may require modest customization steps; (2) our platform is based on high-content imaging, and neurophysiological function must be assessed using separate methodology, likely at lower throughput; (3) similarly, reliance on SYNAPSIN1 on MAP2 as a readout for presynaptic density in developing neurons may be capturing sites that do not then develop into fully functional/mature synapses with postsynaptic partners; (4) while the timepoints of manipulations and analyses can be altered in our assay, longitudinal studies in the same cells are not possible unless suitable genetically encoded fluorescent markers are employed; and (5) our platform is designed for 2D culture to allow for precise control over cell densities and ratios but does not fully reflect the structural complexity present in 3D systems (e.g., spheroids, organoids).

# Materials and methods

## Key resources table

| Reagent type (species) or resource | Designation | Source or reference | Identifiers | Additional information |
|---|---|---|---|---|
| cell line (*Homo sapiens*) | WA01 | WiCell Research Institute | H1 | *Thomson et al., 1998* |
| cell line (*Homo sapiens*) | Primary astrocytes | ScienCell Research Laboratories | Cat#1800 | |
| transfected construct (*Homo sapiens*) | SYNAPSIN1 and control siRNA | Accell Dharmacon | Cat#A-12362-16-0005 and Cat#K-005000-R1-01 | transfected construct (*Homo sapiens*) |

*Continued on next page*

*Continued*

| Reagent type (species) or resource | Designation | Source or reference | Identifiers | Additional information |
|---|---|---|---|---|
| antibody | Anti-human SYNAPSIN1 (Rabbit polyclonal) | Millipore | Cat#AB1543 RRID: AB_2200400 | IF (1:1000) WB (1:1000) |
| antibody | Anti-human MAP2 (Chicken polyclonal) | Abcam | Cat#ab5392 RRID: AB_2138153 | IF (1:1000) |
| antibody | Anti-human SYNAPTOPHYSIN (*Mouse monoclonal*) | Cell Signaling Technology | Cat#9020 RRID: AB_2631095 | IF (1:500) |
| antibody | Anti-human PSD95 (*Mouse monoclonal*) | Thermo Fisher Scientific | Cat#MA1-046 RRID: AB_2092361 | IF (1:500) |
| antibody | Anti-human GAPDH (*Mouse monoclonal*) | Millipore | Cat#MAB374 RRID: AB_2107445 | WB (1:1000) |
| antibody | Anti-human HOMER (Rabbit polyclonal) | Synaptic Systems | Cat#160003 RRID: AB_887730 | WB (1:500) |
| antibody | Anti-human BAIAP2 (Rabbit polyclonal) | Thermo Fisher Scientific | Cat#PA5-30386 RRID: AB_2547860 | WB (1:500) |
| antibody | Anti-human COFILIN | Abcam | Cat#ab42824 RRID: AB_879739 | WB (1:1000) |
| recombinant DNA reagent | pAAVS1-iNGN2-Zeo (plasmid) | This study | | |
| sequence-based reagent | Forward NGN2 | This study | PCR primers | AGGAAATGGGGGTGTGTCAC |
| sequence-based reagent | Reverse NGN2 | This study | PCR primers | GAGCTCCTCTGGCGATTCTC |
| commercial assay or kit | DNeasy Blood and tissue kit | Qiagen | Cat#695004 | |
| commercial assay or kit | RLTplus Lysis buffer | Qiagen | Cat#1053393 | |
| commercial assay or kit | RNeasy micro/mini kit | Qiagen | Cat#74034 | |
| chemical compound, drug | Geneticin | Life Technologies | Cat#10131035 | 50 µg.mL$^{-1}$ |
| chemical compound, drug | SB431542 | Tocris | Cat#1614 | 10 µM; 5 µM |
| chemical compound, drug | XAV939 | Stemgent | Cat#04–00046 | 2 µM; 1 µM |
| chemical compound, drug | LDN-193189 | Stemgent | Cat#04–0074 | 100 nM; 50 nM |
| chemical compound, drug | Doxycycline hyclate | SIgma | Cat#D9891 | 2 µg.mL$^{-1}$ |
| chemical compound, drug | Y27632 | Stemgent | Cat#04–0012 | 5 mM |
| chemical compound, drug | Zeocin | Invitrogen | Cat#46–059 | 1 µg.mL$^{-1}$ |
| chemical compound, drug | BDNF | R&D Systems | Cat#248-BD/CF | 10 ng.mL$^{-1}$ |
| chemical compound, drug | CTNF | R&D Systems | Cat#257-NT-CF | 10 ng.mL$^{-1}$ |
| chemical compound, drug | GDNF | R&D Systems | Cat#212-GD/CF | 10 ng.mL$^{-1}$ |
| chemical compound, drug | Floxuridine | Sigma-Aldrich | Cat#F0503-100MG | 10 µg.mL$^{-1}$ |

*Continued on next page*

*Continued*

| Reagent type (species) or resource | Designation | Source or reference | Identifiers | Additional information |
|---|---|---|---|---|
| chemical compound, drug | Compound library | SelleckChem | L3500 | |
| software, algorithm | Columbus Acapella | Perkin Elmer | | |
| software, algorithm | CellProfiler | The Broad Institute | | |

## Stem cell culture

The XY human ESC line WA01 (H1) and the XY human iPSC line DS2U were commercially obtained from WiCell Research Institute (*Thomson et al., 1998*; *Weick et al., 2013*; https://www.wicell.org/). Stem cell culture was carried out as previously described (*Hazelbaker et al., 2020*; *Hazelbaker et al., 2017*). In brief, stem cells were grown and maintained in StemFlex medium (Gibco, A3349401) on geltrex precoated plates (Life Technologies, A1413301) under standard conditions (37°C, 5% $CO_2$). Cells were passaged using TrypLE Express (Life Technologies, 12604021). All cell lines underwent QC testing to confirm normal karyotypes, absence of mycoplasma, expression of pluripotency markers, and tri-lineage potential. G-band karyotyping analysis was performed by Cell Line Genetics.

## Generation of inducible NGN2 system

TALENs (AAVS1-TALEN-L and AAVS1-TALEN-R; Addgene, 59025/59026; *González et al., 2014*) were used to target the first intron of the constitutively expressed gene *PPP1R12C* at the AAVS1 locus with the pAAVS1-iNGN2-Zeo plasmid containing TetO-NGN2-P2A-Zeo and CAG-rtTA. In brief, hPSCs were dissociated into single-cell suspension with TrypLE (Gibco, 12604-021). $2.5 \times 10^6$ cells were resuspended in 120 μL of R Buffer (Thermo Fisher Scientific, MPK10096) and mixed with 1.5 μg each of AAVS1-TALEN-L and AAVS1-TALEN-R, and 10 μg of pAAVS1-iNGN2-Zeo plasmid. Cells were electroporated using the Neon Transfection electroporation system (Thermo Fisher Scientific, MPK10096) at 1050 V, 30 ms, and 2 pulses, and plated on a 10 cm plate. 24 hr after electroporation and indefinitely, the cells were selected with geneticin (50 μg.mL$^{-1}$, Life Technologies, 10131035). At the fifth day of selection, $2 \times 10^4$ cells were plated on a 10 cm plate for clonal selection. After colony formation, colonies were picked and transferred to a 96-well plate for genomic DNA extraction and PCR analysis of plasmid integration.

## Genomic DNA isolation and genotyping PCR

Genomic DNA (gDNA) from the iNGN2-H1 cell line was extracted from hPSCs with the DNeasy Blood and Tissue kit according to the manufacturer's instructions (QIAGEN, 69504). PCR of the gDNA was performed with the primer pair forward 5′-AGGAAATGGGGGGTGTGTCAC-3′ (in the AAVS1 locus) and reverse 5′- GAGCTCCTCTGGCGATTCTC-3′ (in the NGN2 DNA sequence).

## Human neuron generation

Human neurons were generated as previously described (*Nehme et al., 2018*; *Zhang et al., 2013*). In brief, on day 0, hPSCs were differentiated in N2 medium (500 mL DMEM/F12 [1:1] [Gibco, 11320-033]), 5 mL Glutamax (Gibco, 35050-061), 7.5 mL sucrose (20%, Sigma, S0389), 5 mL N2 supplement B (StemCell Technologies, 07156) supplemented with SB431542 (10 μM, Tocris, 1614), XAV939 (2 μM, Stemgent, 04-00046), and LDN-193189 (100 nM, Stemgent, 04-0074) along with doxycycline hyclate (2 μg.mL$^{-1}$, Sigma, D9891) and Y27632 (5 mM, Stemgent 04-0012). Day 1 was a step-down of small molecules, where N2 medium was supplemented with SB431542 (5 μM, Tocris, 1614), XAV939 (1 μM, Stemgent, 04-00046), and LDN-193189 (50 nM, Stemgent, 04-0074) with doxycycline hyclate (2 μg.mL$^{-1}$, Sigma, D9891) and Zeocin (1 μg.mL$^{-1}$, Invitrogen, 46-059). On day 2, N2 medium was supplemented with doxycycline hyclate (2 μg.mL$^{-1}$, Sigma, D9891) and Zeocin (1 μg.mL$^{-1}$, Invitrogen, 46-059). Starting on day 3, cells were maintained in Neurobasal media (500 mL Neurobasal [Gibco, 21103-049], 5 mL Glutamax [Gibco, 35050-061], 7.5 mL Sucrose [20%, Sigma, S0389], 2.5 mL NEAA [Corning, 25-0250 Cl]) supplemented with B27 (50x, Gibco, 17504-044), BDNF, CTNF, GDNF (10 ng.mL$^{-1}$, R&D Systems 248-BD/CF, 257-NT/CF, and 212-GD/CF) and doxycycline hyclate (2 μg.mL$^{-1}$,

Sigma, D9891). From day 4 to day 5, Neurobasal media was complemented with the antiproliferative agent floxuridine (10 μg.mL⁻¹, Sigma-Aldrich, F0503-100MG).

## Human primary astrocytes

Human primary cortical astrocytes (hpAs) were obtained from ScienCell Research Laboratories (1800) and cultured according to the manufacturer's instructions.

## mRNA sequencing and analysis

Three to six biological replicates of hN + hpA co-cultures per condition (72 hr DMSO treated, 72 hr (+)-JQ1 treated, 72 hr Birabresib treated) were harvested in RLTplus Lysis buffer (QIAGEN, 1053393). Total RNA was isolated using the RNeasy micro/mini plus kit (QIAGEN, 74034). Libraries were prepared using Roche Kapa mRNA HyperPrep strand-specific sample preparation kits from 200 ng of purified total RNA according to the manufacturer's protocol using a Beckman Coulter Biomek i7. The finished dsDNA libraries were quantified by Qubit fluorometer and Agilent TapeStation 4200. Uniquely dual-indexed libraries were pooled in equimolar ratio and subjected to shallow sequencing on an Illumina MiSeq to evaluate library quality and pooling balance. The final pool was sequenced on an Illumina NovaSeq 6000 targeting 30 million 100 bp read pairs per library. Sequenced reads were aligned to the UCSC hg19 reference genome assembly and gene counts were quantified using STAR (v2.7.3a) (*Dobin et al., 2013*). Differential gene expression testing was performed by DESeq2 (v1.22.1) (*Love et al., 2014*). RNAseq analysis was performed using the VIPER snakemake pipeline (*Cornwell et al., 2018*). Library preparation, Illumina sequencing, and VIPER workflow were performed by the Dana-Farber Cancer Institute Molecular Biology Core Facilities.

## Automated cell plating

hNs and hpAs were harvested with Accutase (Innovative Cell Technology, Inc, AT104-500), quenched in Neurobasal media, spun 5 min at 1000 rpm at room temperature (RT), passed through a 40 μm filter, and counted using the Countess Automated Cell Counter (Thermo Fisher Scientific, AMQAX1000). Cells were mixed in NBM media to reach a seeding density of 40,000 neurons.cm⁻² and 100,000 astrocytes.cm⁻² per well. A liquid handling dispenser (Personal Pipettor, ApricotDesigns) was used to uniformly plate the cell mixture in the geltrex-coated 60-inner wells of 96-well plates (PerkinElmer, CellCarrier-96, 6005558).

## Small molecule dilution and addition

The Selleck library (L3500) consisted of 8 × 96-well plates containing 376 small molecules at 10 mM in DMSO (100%). On the day of small molecule addition, the relevant plate was diluted with an automatic liquid handling dispenser (ApricotDesigns, Personal Pipettor). Each diluted plate was screened on the same day, with each small molecule tested against the cell type of interest in three different 96-well plates (in triplicate) at 3 μM.

## siRNA-mediated knockdown

Human SYNAPSIN1 and control siRNA (Accell Dharmacon, A-12362-16-0005 and K-005000-R1-01) were aliquoted (100 μM) and stored at –20°C according to the manufacturer's recommendations. siRNA (1 μM) in Accell Delivery Media (Accell Dharmacon, B-002222-UB-100) was added to human co-cultures on day 18 of hN differentiation for 72 hr followed by fixation and analysis. Normality was confirmed using the Kolmogorov–Smirnov test.

## Immunocytochemistry

Immunofluorescence was performed using an automatic liquid handling dispenser (ApricotDesigns, Personal Pipettor). Cells were washed abundantly in 1× PBS, fixed for 20 min in PFA (4%, Electron Microscopy Sciences, 15714S) plus Sucrose (4%, Sigma, S0389), washed abundantly in 1× PBS, permeabilized and blocked for 20 min in horse serum (4%, Thermo Fisher, 16050114), Triton X-100 (0.3%, Sigma, T9284), and glycine (0.1 M, Sigma, G7126) in 1× PBS. Primary antibodies were then applied at 4°C overnight in 1× PBS supplemented with horse serum (4%, Thermo Fisher, 16050114). The following synaptic antibodies were used: rabbit anti-human SYNAPSIN1 (1:1000, Millipore, AB1543), chicken anti-human MAP2 (1:1000, Abcam, ab5392), mouse anti-synaptophysin (1:500; 7H12 Cell

Signaling Technology, 9020), and mouse anti-PSD-95 (1:500 7E3-1B8 Invitrogen, MA1-046). After abundant washes with Triton X-100 (0.3%, Sigma, T9284) in 1× PBS, cells were exposed for 1 hr at RT to secondary antibodies: goat anti-chicken Alexa Fluor 488 (1:1000, Thermo Fisher, A21131), donkey anti-rabbit Alexa Fluor 555 (1:1000, Thermo Fisher, A31572), goat anti-mouse Alexa Fluor 647 (1:1000, Thermo Fisher), as well as DAPI (1:5000, Thermo Fisher Scientific, D1306) and TrueBlack (1:5000, Biotium, 23007) in 1× PBS supplemented with horse serum (4%, Thermo Fisher, 16050114). Finally, cells were abundantly washed with 1× PBS and stored at 4°C.

## Immunoblotting

Cell lysates were made from iNGN2-H1 hNs monocultures or from hN + hpA co-cultures (40,000 neurons.cm$^{-2}$ and 100,000 astrocytes.cm$^{-2}$ per well) on 6-well plates. Medium was refreshed once per week. Cells were treated with the selected small molecules at the optimal concentration for 72 hr starting on day 18 of hN differentiation. Cells were abundantly washed with PBS 1× before lysis in RIPA buffer (Life Technologies, 89901) supplemented with protease and phosphatase inhibitor (Thermo Scientific, 88669). Before blotting, samples were triturated with repetitive up and down using U-100 Insulin Syringe (B-D, 329461), centrifuged at 10,000 × *g* for 15 min at 4°C, and the supernatants were collected. Proteins were denatured by boiling at 95°C for 10 min and subjected to a Pierce BCA protein assay (Life Technologies, 23227) for determination of the protein concentration. For each sample, the same amount of total protein extracted (5–10 µg) was loaded and separated by SDS-PAGE. The proteins were then transferred to a PVDF membrane (Bio-Rad, 1704156), blocked in 5% Difco Skim Milk (BD, 232100) in TBS-Tween 20 (1/1000; Sigma-Aldrich, T5912-1L and P9416-50ML) before blotting with the primary antibodies: rabbit anti-human SYNAPSIN1 (1:1000, Millipore, AB1543), rabbit anti-human Homer (1/500, Synaptic Systems, 160003), rabbit anti-human BAIAP2 (1/500, Thermo Fisher Scientific, PA5-30386), and mouse anti-human GAPDH (1:1000, Millipore, MAB374). For visualization, donkey anti-mouse IRDye 800CW (1:5000, LI-COR, 926-32212) and donkey anti-rabbit IRDye 680RD (1:5000, LI-COR, 926-68073) were used before detection with Odyssey DLx imaging system (LI-COR, 9140).

## Image acquisition and analysis paradigms
### Plating consistency

24 hr after co-culture, a random 96-well-plate was fixed and stained. 2 channels (DAPI, MAP2), 4 stacks.field$^{-1}$, each stack separated by 0.5 µm, 25 fields.well$^{-1}$, 60 wells.96-well plate$^{-1}$ were acquired at ×20 (Plan Apo $\lambda$, NA 0.75, air objective, Nikon) with a high-content screening confocal microscope (ImageXpress Micro 4, Molecular Devices). Acquired images were transferred and stored on a Columbus (PerkinElmer) server. A Columbus Acapella software (PerkinElmer) algorithm was designed to discriminate hN nuclei from hpA nuclei. DAPI and MAP2 channels of each field were merged in a single plane to create maximum projection images. DAPI projected images were filtered using a Gaussian method and the nuclei population was detected based on the C method (diameter >20 µm). Intensity and morphological properties of each identified nuclei were calculated. Among all nuclei, the hN nuclei were identified according to their intensity, contrast, as well as area, roundness, and distinct MAP2-positive soma (*Figure 1—figure supplement 2*). This script allowed quantification of the mean number (and the standard deviation) of hNs identified in each well, and the coefficient of variance (%) per plate. Thresholding for plating consistency was set above 4000 hNs per well and a covariance below 8% per plate.

### Synapse detection

Three channels (DAPI, MAP2, SYNAPSIN1), 5–8 stacks.field$^{-1}$ at 0.3 µm distance, 12 fields.well$^{-1}$, 60 wells.96-well plate$^{-1}$ were acquired with a ×20 objective (NA 1.0, water objective) with a high-content screening confocal microscope (Opera Phenix, PerkinElmer). Image analysis pipelines were built using the open-source CellProfiler 3.1.5 software (https://www.cellprofiler.org/; *Carpenter et al., 2006*; RRID:SCR_007358). Each of the 12 fields per well were analyzed independently. The first pipeline was designed to merge in a single plane (maximum projection) the 5–8 stacks of the raw image for each channel. The output files generated by the first pipeline were the input files for a second pipeline. The second pipeline was developed to report different features in MAP2 and SYNAPSIN1 channels. First, translational misalignment was corrected by maximizing the mutual information of the DAPI and

MAP2 channels. The same alignment measurements obtained from the first two input images were applied for the SYNAPSIN1 channel. Illumination correction functions were created by averaging each pixel intensity of all images from each channel across each plate then smoothing the resulting image with a Gaussian filter and a large filter size of $100 \times 100$ pixels. A rescale intensity function was used to stretch each image of each channel to the full intensity range (so that the minimum intensity value had an intensity of zero and the maximum had an intensity of 1). The MAP2 pixel intensities were enhanced by a tophat Tubeness filter with a scale of 2 sigma of the Gaussian. Once the lower and upper bounds of the three-class Otsu thresholding method were set for the MAP2 pixel detection, the resulting images were converted into segmented objects. The SYNAPSIN1 pixels were enhanced using a tophat Speckles filter, thresholded using a two-class Otsu method, processed for SYNAPSIN1 puncta identification within an equivalent diameter range of 1–6 pixels, and de-clumped before being converted into segmented objects. A colocalization module then assigned the relationships between the identified SYNAPSIN1 puncta contained within or only partly touching the MAP2 objects. Finally, the area and the number of the synaptic objects were measured and exported to a *.csv spreadsheet. The CellProfiler pipelines are available at https://github.com/mberryer/ALPAQAS (copy archived at *Berryer, 2023*).

## Data analysis

### Quality control

The output CellProfiler *.csv files were imported in Genedata Screener. As the *.csv files were loaded, a parameterized parser conducted quality control. The mean and standard deviation of the area covered by MAP2 pixels were calculated across all fields in an inter-plate basis. Fields in which the area covered by MAP2 pixels was outside the range of the inter-plate mean ± standard deviation values were removed. Wells containing fewer than five fields were also excluded. The area and the number of the synaptic objects of the remaining fields were averaged and condensed to well-level before being imported into Genedata Screener to perform relevant calculations.

### Synaptic measurements

We analyzed the area occupied by MAP2-positive neurites and the number, area, and density of SYNAPSIN1 puncta localized on MAP2-positive neurites (respectively AreaOccupied and Count of MAP2PositiveNeurites and SYNAPSIN1PunctaOnMAP2PositiveNeurites in the ALPAQAS pipeline). On a well-basis, density was defined as the number of SYNAPSIN1-positive puncta colocalized on MAP2-positive pixels divided by the area covered by the MAP2 positive pixels. The mean values for number and density were calculated for each well and expressed as a percentage of the intra-plate DMSO control-treated wells. Computed on an intra-plate basis, the Z-score for the value X was defined as the difference between X and the mean value of all DMSO-treated wells, divided by the standard deviation value of all DMSO-treated wells.

### SynGO analysis

Individual upregulated DEGs shared between (+)-JQ1 and Birabresib treatment conditions ($p_{adj} < 0.05$ and $\log_2 FC \geq 1$) were used as input with default settings. All genes with average counts $\geq 1$ in the 0.1% DMSO control conditions were used as the background list (for a total of 11,704 genes).

### Power calculations

Cohen's d was calculated for each compound compared to DMSO using pooled standard deviation. The statistical power of each compound was calculated from Cohen's d, number of fields/wells, with alpha 0.05, hypothesis as specified ('two-sided,' 'greater,' or 'less') using the function pwr.norm.test from R package pwr (1.3–0). The number of fields or wells was calculated for increasing Cohen's d and power, with alpha 0.05, using the function pwr.norm.test from R package pwr (1.3–0). Power plots were generated with ggplot2 (3.3.5).

## Statistical analyses

All experiments were performed in three biological replicates and three technical replicates, except the small molecule validation experiments were performed in two biological replicates and eight

technical replicates and the siRNA experiment was performed in one biological replicate and six technical replicates. Biological replicates refer to independent batches of cells/differentiations, and technical replicates refer to independent wells. For the dose–response, the small molecule validation, the hN monoculture experiments, and the immunoblot quantification, significance was assessed using one-way ANOVA with Dunnett's multiple comparisons test; *p<0.05, **p<0.01, ***p<0.001. In the hN + hpA co-culture versus hN monoculture experiment, the quantification of the density and the area of the individual SYNAPSIN1 puncta and the area covered by MAP2-positive neurites, the significance was determined by a Kolmogorov–Smirnov unpaired $t$-test, ***p<0.001. For mRNA-seq analyses, we used an adjusted p-value cutoff of 0.05 and a $\log_2$fold change cutoff of ±1. p-Values (or adjusted p-values where relevant) 0.05 were considered statistically significant.

## Acknowledgements

We thank members of the Barrett and Rubin labs for insightful discussions and critical reading of the manuscript. We appreciate the support from Ralda Nehme for her useful inputs on the neuronal induction protocol, Maria Alimova and Patrick Byrne at CDoT for their high-throughput imaging expertise, Sam Bryant and Rachel Fox at the Stanley Center for their analytical insights, Jessica Moffitt for technical support, and Francesca Rapino and Maura Charlton for their wisdom and guidance throughout the project. We also thank Ajamete Kaykas and Katie Worringer for the NGN2 AAVS1 targeting construct. We appreciate the analytical support from Zach Herbert at the Dana-Farber Cancer Institute Molecular Biology Core Facilities. This work was supported by a Broad*next*10 grant to LEB and LLR, as well as support from the Stanley Center for Psychiatric Research. KWK, BAC, and AEC were supported in part by a grant from the National Institutes of Health NIH R35 GM122547 to AEC.

## Additional information

### Competing interests

Lee L Rubin: Reviewing editor, eLife. LLR is a founder of Elevian, Rejuveron, and Vesalius Therapeutics, a member of their scientific advisory boards and a private equity shareholder. All are interested in formulating approaches intended to treat diseases of the nervous system and other tissues. He is also on the advisory board of Alkahest, a Grifols company, focused on the plasma proteome and brain aging. None of these companies provided any financial support for the work in this paper. The other authors declare that no competing interests exist.

### Funding

| Funder | Grant reference number | Author |
| --- | --- | --- |
| Stanley Center for Psychiatric Research at the Broad Institute | Broadnext10 | Lee L Rubin Lindy E Barrett |
| National Institutes of Health | R35 GM122547 | Anne E Carpenter |

The funders had no role in study design, data collection and interpretation, or the decision to submit the work for publication.

### Author contributions

Martin H Berryer, Conceptualization, Formal analysis, Investigation, Visualization, Methodology, Writing – original draft, Writing – review and editing, Performed stem cell, neuron and astrocyte cultures, immunostaining, development, optimization and performance of the assays and analyses, and helped to develop CellProfiler pipelines; Gizem Rizki, Conceptualization, Supervision, Methodology, Provided strategic and technical assistance with screening experiments; Anna Nathanson, Data curation, Investigation, Performed protein expression experiments and analyses; Jenny A Klein, Formal analysis, Investigation, Performed immunostaining and analyses; Darina Trendafilova, Data curation, Investigation, Assisted with cell culture and performed protein expression and immunostaining experiments; Sara G Susco, Resources, Generated TALEN edited cell lines; Daisy Lam,

Resources, Generated TALEN edited cell lines; Angelica Messana, Resources, Generated TALEN edited cell lines; Kristina M Holton, Formal analysis, Performed power calculations; Kyle W Karhohs, Software, Methodology, Developed CellProfiler pipelines; Beth A Cimini, Supervision, Methodology, Writing – review and editing, Developed CellProfiler pipelines; Kathleen Pfaff, Supervision, Methodology, Provided strategic and technical assistance with screening experiments; Anne E Carpenter, Supervision, Funding acquisition, Writing – review and editing; Lee L Rubin, Conceptualization, Supervision, Funding acquisition, Writing – review and editing; Lindy E Barrett, Conceptualization, Supervision, Funding acquisition, Writing – original draft, Writing – review and editing

### Author ORCIDs
Beth A Cimini ⓘ http://orcid.org/0000-0001-9640-9318
Anne E Carpenter ⓘ http://orcid.org/0000-0003-1555-8261
Lee L Rubin ⓘ http://orcid.org/0000-0002-8658-841X
Lindy E Barrett ⓘ http://orcid.org/0000-0002-3229-4774

### Decision letter and Author response
Decision letter https://doi.org/10.7554/eLife.80168.sa1
Author response https://doi.org/10.7554/eLife.80168.sa2

## Additional files

### Supplementary files
• Supplementary file 1. List of the 376 small molecules used in the primary screen including molecule name, synonyms, library, supplier, catalog number, target, additional notes, and pathway.

• Supplementary file 2. Power calculations for the primary screen overall, the primary screen by compound, co-cultures at the field level, co-cultures at the well level, neuron monocultures at the field level, and neuron monocultures at the well level.

• Supplementary file 3. Transcriptional analyses after (+)-JQ1 inhibition.

• Supplementary file 4. Transcriptional analyses after Birabresib inhibition.

• Supplementary file 5. SynGO analyses including the GO term ID, domain, name, and p-value.

• MDAR checklist

### Data availability
Data, resource and code availability: All data is available in the manuscript or the supplemental materials. The analysis codes generated in this study are available from GitHub at: https://github.com/mberryer/ALPAQAS (copy archived at *Berryer, 2023*). Engineered cell lines are available upon request and following appropriate institutional guidelines for their use and distribution.

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
