## [Editor Report]

Berryer et al. report on an automated and quantitative platform to study the number of synaptic inputs formed in networks of human excitatory neurons and astrocytes in vitro. The utility of the platform was tested by screening a large collection of small molecules; several modulators of synapse density were identified and validated in follow-up experiments. The automated platform substantially extends what is currently available, particularly with respect to the automation of the initial analysis steps. The positive hits identified here, the inhibitors of bromodomain and extraterminal (BET) family of gene expression regulators, are important, and will likely contribute to the understanding of the mechanisms of human synapse assembly.

---

## [Decision Letter]

**Decision letter after peer review:**

Thank you for submitting your article "An automated high-content synaptic phenotyping platform in human neurons and astrocytes reveals a role for BET proteins in synapse assembly" for consideration by *eLife*. Your article has been reviewed by 3 peer reviewers, one of whom is a member of our Board of Reviewing Editors, and the evaluation has been overseen by Lu Chen as the Senior Editor. The following individuals involved in the review of your submission have agreed to reveal their identity: Matthijs Verhage (Reviewer #2); Carlo Sala (Reviewer #3).

Essential revisions:

All three reviewers have acknowledged the high potential of the automated platform in drug screening and related studies that target human synapses and the novelty of identifying BET inhibitors in affecting synapse density. Some shortcomings of the study have been raised, however. These relate to the limitations of using synapsin I as the sole synaptic marker and the fact that the manuscript could be considerably strengthened by providing further evidence in support of the synaptic actions of BET inhibitors and including statistical analysis to clarify and place limits on the power of the phenotyping platform in detecting small effects. To this end, the following essential revisions are requested, the first two of which require additional experiments. Furthermore, the authors are also requested to address all the points raised by each reviewer.

1) The use of synapsin1 as a sole synapse marker should be experimentally validated in two ways:

(a) In a sample set of experiments, demonstrate the extent to which postsynaptic markers are found apposed to synapsin I and that the distribution of postsynaptic markers match that of synapsin I.

(b) In a sample set of experiments, demonstrate the extent to which other presynaptic vesicle marker and/or active zone markers colocalize with synapsin I and that the distribution of other presynaptic markers match that of synapsin I.

2) The effects of BET inhibitor treatment on the expression of synaptic proteins should be confirmed by Western blots.

3) The synaptic gene expression programs boosted by BET inhibitors should be further characterized with respect to their pre/post expression loci or function (cf. Reviewer 2, major point 3).

4) Further statistical analysis should be performed to clarify the power of the platform in detecting the effects of a certain size (cf. Reviewer 2, major point 1).

*Reviewer #1 (Recommendations for the authors):*

Here the synaptic phenotyping involves three parameters: (i) density of a synapse marker protein, (ii) dendrite area, and (iii) number of viable cells. In particular, the key parameter, synapse density, is reliant on the density of a presynaptic phosphoprotein, synapsin I, whose synaptic function, at least in rodents, might be limited and whose confinement at presynaptic boutons is activity-dependent. Therefore, it is possible that the use of another synaptic marker (presynaptic or postsynaptic) may give different outcomes. In other words, the robustness of a platform based on synapsin 1 as the only synaptic marker is not entirely clear. Providing some experimental support to this end would make the manuscript more convincing (cf. specific points 1, 5).

1) What percentage of Synapsin puncta present on MAP2 are bona fide synapses in that they contain postsynaptic markers? While it may not be necessary to perform the double labelling each time, it would be helpful to know how accurately the readout of presynaptic maker density corresponds to the actual synapse density.

2) Figure 1e, f, and Lines 196-197: The claimed consistency of plating across batches of differentiation needs to be better substantiated. For instance, in panel e, there is a big difference between batches 3 and 4. What is the permissible range for the said consistency and the basis for choosing such criteria warrants an explanation?

3) Figure 3. In order to facilitate a direct comparison of the dosage effects of the drugs on synapsin I density, MAP2 neurite area, and cell survival, the x-axis should be the same across the three parameters.

4) Figures4, 5. The presence of astrocytes in promoting synapsin 1 density on MAP2 neurites is far more potent than the effects of small molecule inhibitors in facilitating synapsin 1 density. Moreover, the presence of astrocytes actually seems to reduce the MAP2 positive neurite area, which will further boost the synapsin 1 density measurement that is made relative to MAP2 neurite area. Could one exclude the possibility that an apparent lack of an increase by the small molecule inhibitors in the absence of astrocytes was because of sensitivity issue?

5) Figure 6. A direct demonstration of an increase in neurexin 3 and/or homer 1 protein levels by immunofluorescence labelling and western blots in the co-culture system used here following BET inhibitor incubation would further strengthen the conclusion of a role played by BET proteins in attenuating synapse formation.

*Reviewer #2 (Recommendations for the authors):*

1. The authors claim they "established a novel, automated, high-content synaptic platform" (line 402). While the extent of automation certainly reaches beyond what has been published, other aspects are not 'novel', especially the central concept of evaluating the density of synapsin1 puncta over MAP2 positive dendrites. It would be good to define a bit more precise the exact step beyond the state of the art, which lies mostly in the automation of plating etc, and to give credit to previous studies that have delivered open-source methods for automated synapse quantifications:

SynQuant https://doi.org/10.1093/bioinformatics/btz760,

SynapseJ https://doi.org/10.3389/fncir.2021.731333,

SynPAnal https://doi.org/10.1371/journal.pone.0115298,

SynD https://doi.org/10.1016/j.jneumeth.2010.12.011

2. For Figures2 and 5, it is more informative to present the real images instead of the binary masks.

3. In line 192 the authors explained one of their quality control tests, it would be good to mention that it is performed 6 days post-plating. As for now, the only indication is in Fig1d.

4. In Figure 1: it is not clear that the staining is performed at DIV21. Swapping panels c and d would probably improve readability? Why panel g shows the dispersions of the data points, while panel e does not? The data are plotted as mean {plus minus} SEM, were the data tested for normality? It is not clear from the method section.

5. Please clarify: In line 267 there are listed 17 small molecules. However, in Figure 2 there are 20 molecules labeled, and the number of labeled molecules for the inhibitors family does not match with the text (e.g., 3 BET inhibitors in the text, but only two labeled).

6. It is not clear from the figures, nor the text in lines 305-320 that the experiments with and without astrocytes are done in parallel.

7. What is the size of the Speckles filter used for the synapse thresholding? The range of 1 to 6 pixels per synapse seems a bit broad, especially in the low-end (1-2 pxs) it might be prone to detect scatter noise. Did the authors try to evaluate how changing those parameters affects detection, and if so, please explain the choice of the declared parameters?

*Reviewer #3 (Recommendations for the authors):*

The present study provides a quantitative platform to count synapses in cultured human neurons derived from iPSCs. This platform was used to identify novel signaling pathways important for synapse formation.

To strengthen the impact of these findings the author should address the following points:

1) All the methodology was used to quantify a presynaptic marker and indeed the effect of BET was measured on presynaptic synapsin1. So it will be nice to show if BET treatment is able to increase the staining of a postsynaptic marker (Homer1 for example), if feasible.

2) The mRNA-seq analysis indicates that BET treatment increases the expression of some synaptic and non synaptic proteins. However, these specific genetic modifications should be confirmed also by WB analysis of some of the identified major proteins.

---

## [Author Response]

Essential revisions:All three reviewers have acknowledged the high potential of the automated platform in drug screening and related studies that target human synapses and the novelty of identifying BET inhibitors in affecting synapse density. Some shortcomings of the study have been raised, however. These relate to the limitations of using synapsin I as the sole synaptic marker and the fact that the manuscript could be considerably strengthened by providing further evidence in support of the synaptic actions of BET inhibitors and including statistical analysis to clarify and place limits on the power of the phenotyping platform in detecting small effects. To this end, the following essential revisions are requested, the first two of which require additional experiments. Furthermore, the authors are also requested to address all the points raised by each reviewer.1) The use of synapsin1 as a sole synapse marker should be experimentally validated in two ways:(a) In a sample set of experiments, demonstrate the extent to which postsynaptic markers are found apposed to synapsin I and that the distribution of postsynaptic markers match that of synapsin I.(b) In a sample set of experiments, demonstrate the extent to which other presynaptic vesicle marker and/or active zone markers colocalize with synapsin I and that the distribution of other presynaptic markers match that of synapsin I.

We tested conditions for a set of two additional presynaptic and four additional post-synaptic antibodies to assess their distribution and colocalization with Synapsin1 on MAP2. Of these antibodies, we obtained quantifiable signal for one additional presynaptic marker (synaptophysin) and one additional postsynaptic marker (PSD-95). Co-localization analyses between synaptophysin and syanapsin1 on MAP2, as well as PSD-95 and synapsin1 on MAP2 are now presented in Figure 1—figure supplement 1 and discussed in detail below (Reviewer #1 point #1 and Reviewer #2 point #2)

2) The effects of BET inhibitor treatment on the expression of synaptic proteins should be confirmed by Western blots.

We have now confirmed that BETi treatment increases protein expression levels of postsynaptic proteins by Western blot analysis, including BAIAP2 and Homer1. Specifically, we show that both JQ1 treatment and Birabresib treatment significantly increase the expression of both postsynaptic scaffolding proteins compared to DMSO treated control. These data are now shown in Figure 6 and discussed in detail below (Reviewer #1 point #5).

3) The synaptic gene expression programs boosted by BET inhibitors should be further characterized with respect to their pre/post expression loci or function (cf. Reviewer 2, major point 3).

As discussed in detail below (Reviewer #2, point 3), we have now added SynGO analysis to Figure 6 and Supplementary File 5 to further characterize synaptic gene expression enhanced by BETi.

4) Further statistical analysis should be performed to clarify the power of the platform in detecting the effects of a certain size (cf. Reviewer 2, major point 1).

To clarify the power of the platform in detecting various effect sizes, we have now calculated Cohen’s d for: (1) the primary screen overall as well as for each small molecule included in the primary screen, (2) validation experiments performed in neuron monocultures and (3) validation experiments performed in neuron + astrocyte co-cultures, and these data have been added to Figure 5, Figure 5—figure supplement 1 and Supplementary File 2. We have also added a discussion of how this informs on study design. These analyses are discussed in detail below (Reviewer #2, point #1).

Reviewer #1 (Recommendations for the authors):Here the synaptic phenotyping involves three parameters: (i) density of a synapse marker protein, (ii) dendrite area, and (iii) number of viable cells. In particular, the key parameter, synapse density, is reliant on the density of a presynaptic phosphoprotein, synapsin I, whose synaptic function, at least in rodents, might be limited and whose confinement at presynaptic boutons is activity-dependent. Therefore, it is possible that the use of another synaptic marker (presynaptic or postsynaptic) may give different outcomes. In other words, the robustness of a platform based on synapsin 1 as the only synaptic marker is not entirely clear. Providing some experimental support to this end would make the manuscript more convincing (cf. specific points 1, 5).1) What percentage of Synapsin puncta present on MAP2 are bona fide synapses in that they contain postsynaptic markers? While it may not be necessary to perform the double labelling each time, it would be helpful to know how accurately the readout of presynaptic maker density corresponds to the actual synapse density.

As mentioned above, we tested conditions for four additional post-synaptic antibodies, drawing from those used in published studies of human cellular models (and species that would not cross-react with the antibodies used for Synapsin1 and MAP2). Specifically, we tested antibodies against PSD-95, NLGN4, Homer1 and BAIAP2 at a range of concentrations in co-cultures generated from two independent cell lines. Of these antibodies, we obtained quantifiable signal for PSD-95, while NLGN4, Homer1 and BAIAP2 appeared to be of poor quality in our cellular systems (e.g., nonspecific signal, high signal in astrocytes, etc.).

As shown in Figure 1—figure supplement 1, analysis of PSD-95 revealed that 43.1% of PSD-95 puncta on MAP2 also colocalized with synapsin1, and 28.8% of synapsin1 puncta on MAP2 also colocalized with PSD-95. Discussions of these data and limitations have been significantly elaborated upon on pages 10-11 of the Results section in the PDF and pages 24-25 and 29 of the Discussion section in the PDF. For example, we discuss how the partial colocalization could be due both to the relative immaturity of the synapses (presynaptic assembly preceding postsynaptic assembly at this early stage of neuronal development) as well as the overall poorer quality of the PSD-95 signal in human cellular material (PSD-95 signal was of insufficient quality and consistency for screening applications and was generally quite difficult to resolve as compared to Synapsin1). We include the reliance on Synapsin1 on MAP2 as an explicit limitation. We also add additional references on the use of Synapsin1 to label cortical glutamatergic synapses in rodent (e.g., Micheva 2010) and the use of Synapsin1 on MAP2 as a pan-synaptic marker in human cellular material (e.g., Chanda et al. 2019, Pak et al. 2015, Yi et al. 2016).

2) Figure 1e, f, and Lines 196-197: The claimed consistency of plating across batches of differentiation needs to be better substantiated. For instance, in panel e, there is a big difference between batches 3 and 4. What is the permissible range for the said consistency and the basis for choosing such criteria warrants an explanation?

In brief, we tested a range of hN plating densities based on the published literature of synaptic screens made with rodent neurons. We found that 4,000 hNs/well (as assessed by the number of surviving hNs from a plate randomly selected one day after co-culture) was a minimum for forming a synaptic network and 12,000 hNs/well was a maximum after which point, the hNs began to clump. This information is now indicated in the Figure 1e and in the corresponding text and figure legend. For reference, rodent screens typically plate down between 10,000 – 20,000 neurons/well (surviving number not assessed). It’s important to note the consistency of plating is critical within each batch so that approximately the same number of hNs are treated to allow for comparison. The consistency of plating is based on the standard error to the mean of each batch (Figure 1e) and on the intra-plate co-variance within each batch (Figure 1f). Each compound treatment condition is compared to controls contained within the same plate and the same batch; different batches are considered different experimental replicates where compound treatment is again compared to intra-plate / intra-batch controls. In other words, we are not comparing conditions across batches that have different absolute numbers of neurons. These points have now been elaborated upon in the text (please see pages 12 of the Results section in the PDF).

3) Figure 3. In order to facilitate a direct comparison of the dosage effects of the drugs on synapsin I density, MAP2 neurite area, and cell survival, the x-axis should be the same across the three parameters.

Figure 3 has now been changed to show the same x-axis for Synapsin1 density, MAP2 neurite area and cell survival using histogram format. The dose response curves for each of the 17 compounds have been moved to Figure 3—figure supplement 1. All data on dosage effects are now included both as dose response curves and histogram format (Figure 3, Figure 3—figure supplements 1-2) to facilitate comparison with area covered by MAP2 (histograms in Figure 3—figure supplement 3) and number of DAPI positive nuclei (histograms in Figure 3—figure supplement 4).

4) Figures4, 5. The presence of astrocytes in promoting synapsin 1 density on MAP2 neurites is far more potent than the effects of small molecule inhibitors in facilitating synapsin 1 density. Moreover, the presence of astrocytes actually seems to reduce the MAP2 positive neurite area, which will further boost the synapsin 1 density measurement that is made relative to MAP2 neurite area. Could one exclude the possibility that an apparent lack of an increase by the small molecule inhibitors in the absence of astrocytes was because of sensitivity issue?

As discussed below, to address the issue of sensitivity of our validation assays, we have now performed power calculations for both the neuron monoculture condition and the co-culture condition. These analyses indicate that we are sufficiently powered to detect changes in both assays, given the predicted effect sizes (Figure 5, Figure 5—figure supplement 1 and Supplementary File 2). For example, effect sizes in the co-culture condition were sufficiently large that we could scale down the number of fields (n = 47-80) and wells (n = 6-9) analyzed in future experiments. For the monoculture condition, effect sizes were much smaller; we scaled up the number of fields (n = 404-417) and wells (n = 52) analyzed which was sufficient to achieve 95% power for detecting changes due to Birabresib and I-BET151. While we are unable to exclude a technical limitation in detecting differences in the absence of astrocytes, the point we make in the text is that screens carried out in the absence of astrocytes are likely to miss hits, for either technical or biological reasons. We have now expanded on this point in the text (please see pages 19-20 of the Results section and page 26 of the Discussion section in the PDF).

5) Figure 6. A direct demonstration of an increase in neurexin 3 and/or homer 1 protein levels by immunofluorescence labelling and western blots in the co-culture system used here following BET inhibitor incubation would further strengthen the conclusion of a role played by BET proteins in attenuating synapse formation.

We have now performed western blot analyses for two postsynaptic proteins, Homer1 and BAIAP2, following both JQ1 treatment and birabresib treatment, compared to DMSO treated control. These experiments, which are now included in Figure 6, confirm upregulation of both Homer1 and BAIAP2 protein expression following BETi, supporting our transcript results and strengthening the conclusion that BET proteins can attenuate synapse formation. We also tested additional antibodies for genes increased at the transcript level such as AMPH but did not obtain specific signal in western blot analysis (data not shown). These data are also discussed on page 23 of the text in the PDF. As discussed above, we also tested Homer1 and BAIAP2 for immunofluorescence labeling, but did not obtain specific signal in our cultures for this application.

Reviewer #2 (Recommendations for the authors):1. The authors claim they "established a novel, automated, high-content synaptic platform" (line 402). While the extent of automation certainly reaches beyond what has been published, other aspects are not 'novel', especially the central concept of evaluating the density of synapsin1 puncta over MAP2 positive dendrites. It would be good to define a bit more precise the exact step beyond the state of the art, which lies mostly in the automation of plating etc, and to give credit to previous studies that have delivered open-source methods for automated synapse quantifications:SynQuant https://doi.org/10.1093/bioinformatics/btz760,SynapseJ https://doi.org/10.3389/fncir.2021.731333,SynPAnal https://doi.org/10.1371/journal.pone.0115298,SynD https://doi.org/10.1016/j.jneumeth.2010.12.011

We thank the reviewer for this comment and have now added the above references and more clearly defined aspects of novelty presented in the current study (please see page 25 of the Discussion section in the PDF).

2. For Figures2 and 5, it is more informative to present the real images instead of the binary masks.

We have now included raw images for Figures 2 and 5, and we have also kept the binary masks. We wanted to show individual zoomed in examples (which we agree, look much better with the real images) but also panning out to give a sense of the overall co-culture and MAP2 distribution (which was easier to show clearly with the binary masks).

3. In line 192 the authors explained one of their quality control tests, it would be good to mention that it is performed 6 days post-plating. As for now, the only indication is in Fig1d.

This information has now been specified in the text.

4. In Figure 1: it is not clear that the staining is performed at DIV21. Swapping panels c and d would probably improve readability? Why panel g shows the dispersions of the data points, while panel e does not? The data are plotted as mean {plus minus} SEM, were the data tested for normality? It is not clear from the method section.

We appreciate this comment on readability and have now swapped Figure 1c and Figure 1d, and included the day 21 time-point next to the staining examples and on page 12 of the text. We also now show the dispersion of the individual datapoints in Figure 1e (this includes 60 wells per plate and 60 points per bar). We also tested the data for normality which was confirmed using the Kolmogorov-Smirnov test; this information is included in the Methods section.

5. Please clarify: In line 267 there are listed 17 small molecules. However, in Figure 2 there are 20 molecules labeled, and the number of labeled molecules for the inhibitors family does not match with the text (e.g., 3 BET inhibitors in the text, but only two labeled).

This information has now been clarified in the text (please see pages 15-16 in the PDF). In particular, we followed-up on 17 small molecules but Figure 2 also shows anticarcinogens that decreased synaptic density (as well as viability) and were thus not followed up on with a dose response. We have separated the anticarcinogens in the legend in Figure 2 from the rest of the selected compounds and added these details to the corresponding figure legend. Birabresib was also not in the original compound library but we chose to add it for validation; BETi was an unexpected but particularly interesting hit class so we wanted additional validation of this result using a third independent BETi.

6. It is not clear from the figures, nor the text in lines 305-320 that the experiments with and without astrocytes are done in parallel.

Experiments performed with and without astrocytes were indeed done in parallel and this information has now been clarified in the text (please see page 18 in the PDF).

7. What is the size of the Speckles filter used for the synapse thresholding? The range of 1 to 6 pixels per synapse seems a bit broad, especially in the low-end (1-2 pxs) it might be prone to detect scatter noise. Did the authors try to evaluate how changing those parameters affects detection, and if so, please explain the choice of the declared parameters?

We had previously performed a cumulative distribution of Synapsin1 puncta diameter in pixels, which we have now included in Figure 1—figure supplement 4 and discussed in the text (please see pages 13-14 in the PDF). Of note, while estimates of Synapsin1 puncta diameters vary considerably, several recent studies have focused on diameters at the lower end (in the 1-3 pixel range in our assay). We used a broader distribution to capture Synapsin1 puncta which can be modified by end users in the platform for project specific goals.

Reviewer #3 (Recommendations for the authors):The present study provides a quantitative platform to count synapses in cultured human neurons derived from iPSCs. This platform was used to identify novel signaling pathways important for synapse formation.To strengthen the impact of these findings the author should address the following points:1) All the methodology was used to quantify a presynaptic marker and indeed the effect of BET was measured on presynaptic synapsin1. So it will be nice to show if BET treatment is able to increase the staining of a postsynaptic marker (Homer1 for example), if feasible.

As discussed above, we have now shown that BETi significantly increases the expression of the postsynaptic markers Homer1 and BAIAP2 compared to DMSO treated control by western blot analysis (please see Figure 6). We tested these same antibodies for immunofluorescence labeling in our cultures but unfortunately did not obtain specific signal for that application.

2) The mRNA-seq analysis indicates that BET treatment increases the expression of some synaptic and non synaptic proteins. However, these specific genetic modifications should be confirmed also by WB analysis of some of the identified major proteins.

As discussed above, we have now shown that BETi significantly increases the expression of the postsynaptic markers Homer1 and BAIAP2 compared to DMSO treated control, by western blot analysis. We focused on postsynaptic markers, as these were more challenging to capture in our assay by immunofluorescence but could more readily be captured by western blot analysis; postsynaptic genes were also a category which emerged from our SynGO analyses and thus seemed most relevant to confirm in the RNAseq dataset. We did test antibodies for other genes altered at the RNAseq level (e.g., AMPH) but did not obtain specific signal for western blot analysis.